# WHAT MAKES THE PREFERRED THINKING DIRECTION FOR LLM IN MULTI-CHOICE QUESTIONS?

## ABSTRACT

Language models usually use left-to-right (L2R) autoregressive factorization. However, L2R factorization may not always be the best inductive bias for all tasks. Therefore, we investigate whether alternative factorizations of the text distribution could be beneficial in specific task domains. We investigate right-to-left (R2L) training as a compelling alternative, focusing on multiple-choice questions (MCQs) as a test bed for knowledge extraction and reasoning. Through extensive experiments across various model sizes (2B-8B parameters) and training datasets, we find that R2L models can significantly outperform L2R models on a subset of MCQ benchmarks (4 out of 11 evaluated tasks), including logical reasoning, commonsense understanding, and truthfulness assessment tasks. Our analysis reveals that this domain-specific performance difference may be fundamentally linked to multiple factors including calibration, computability, and directional conditional entropy. We ablate the impact of these factors through controlled simulation studies using arithmetic tasks, where the impacting factors can be better disentangled. Our work demonstrates that the standard assumption of L2R as the universally optimal factorization is not always valid, and that exploring alternative factorizations can lead to task-specific improvements in LLM capabilities. We provide theoretical insights into when each reasoning order might be more advantageous based on the statistical and structural properties of the target distribution.

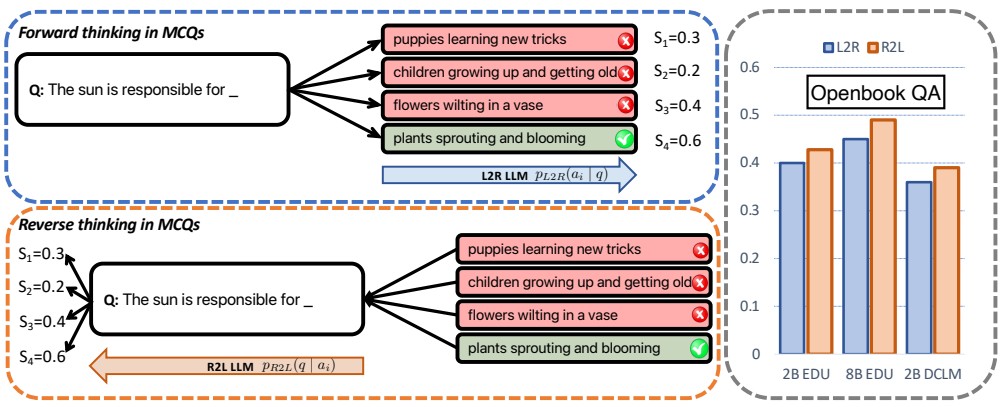

Figure 1: Reverse Thinking in MCQs. **Top left**: Standard forward thinking evaluates each answer choice based on the question and selects the one with the highest relevance score in a L2R LLM. **Bottom left**: Reverse thinking evaluates the question based on each answer choice and selects the answer that maximizes the relevance score in a R2L LLM. **Right**: Reverse thinking consistently outperforms forward thinking in certain MCQ tasks, independent of training data and model size.

## 1 INTRODUCTION

Large Language Model (LLM) pretraining commonly employs left-to-right (L2R) next-token prediction, an approach that enables efficient parallelization and caching. This method models the text

distribution $p(x)$ as a factorized autoregressive chain as $p(x_t|x_{<t})$. L2R naturally aligns with human cognitive processes of text generation and reasoning, making it well-suited for inference tasks.

However, while perfect modeling of each $p(x_t|x_{<t})$ would theoretically enable exact recovery of the data distribution $p(x)$, neural networks inevitably introduce approximation errors for each $p(x_t|x_{<t})$. These errors compound over timestep $t$ during inference, potentially resulting in hallucinations and repetitions in generation (Bengio et al., 2015; Zhang et al., 2023). Further, L2R factorization can result in inductive biases that lead to unwanted behaviors. For example, Allen-Zhu & Li (2023a) show that inverse search is challenging for L2R LLMs, and Berglund et al. (2023) demonstrate the "reversal curse" where models trained on forward text data struggle with inverse relationships.

We investigate whether L2R is optimal, and if alternative factorizations might capture unique aspects of the data distribution that complement L2R. Can specific factorizations achieve lower approximation errors compared to L2R, or reduce L2R's inherent bias in particular task domains?

Autoregressive modeling in right-to-left (R2L) fashion factorizes $p(x)$ as $p(x_t|x_{>t})$, which presents a particularly promising alternative that has been examined in previous work (Papadopoulos et al., 2024; Berglund et al., 2023; Zhang-Li et al., 2024). This setup views the task as predicting the previous token, and it can achieve prediction losses comparable to the L2R next token prediction objective, due to its symmetry to L2R. While R2L may seem counterintuitive given human language processing patterns, it may enable more efficient knowledge extraction in certain scenarios by aligning with the natural direction of information flow in those cases, and it could provide complementary inductive biases that help with specific reasoning tasks.

We investigates three questions: (1) **How to evaluate R2L models on knowledge extraction and basic reasoning tasks?** (2) **Can R2L factorization match or surpass L2R's capabilities in knowledge extraction and reasoning for downstream tasks?** (3) **What are underlying factors determining the preference of L2R or R2L factorizations?** To address these questions, we conducted controlled experiments comparing L2R and R2L models trained with identical data and computational resources. We evaluated both factorization approaches using standard LLM benchmarks with Multiple-Choice Questions (MCQs). For simplicity, we limit our comparison to MCQs, and leave the evaluations for generative tasks as future work. For R2L models, we applied Bayesian inference to implement "reverse thinking," evaluating choices based on their likelihood of generating the prompt (Figure 1).

Our results reveal a surprising and previously unobserved empirical finding: R2L models can significantly outperform L2R models on a subset of standard MCQ benchmarks (4 out of 11 evaluated tasks), including tasks requiring logical reasoning, commonsense understanding, and truthfulness assessment. This finding—where backward thinking can sometimes yield superior performance—is a phenomenon not previously observed or widely recognized in the current literature, especially at the scale and across the diverse benchmarks we evaluated. This observation challenges the prevailing fundamental assumption that L2R autoregressive factorization is universally optimal for all tasks. We emphasize that our primary contribution is not to claim R2L superiority, but to demonstrate that the optimal factorization direction is task-dependent and linked to the statistical and structural properties of the target distribution.

Beyond this empirical discovery, our work introduces new perspectives on analyzing the performance differences between L2R and R2L models by proposing three potential underlying factors: *calibration*, *computability*, and, critically, *conditional entropy*. The role of conditional entropy in explaining the preferred reasoning direction, and generally understanding reasoning machinery, is a novel theoretical insight introduced in this paper. We empirically verify this hypothesis, showing that lower conditional entropy generally correlates with higher accuracy in the reasoning direction. Our primary contribution lies not in demonstrating that one factorization is universally superior, but in developing a principled framework for understanding when and why different factorization directions may be preferred.

Nevertheless, these factors are intricately interwoven in actual MCQs, complicating the analysis. To disentangle these factors and ablate on their impact to the performance of L2R or R2L factorization, we design a controlled simulation study using arithmetic tasks, revealing how various factors influence the effectiveness of certain factorization. Our code and model checkpoints have been made publicly available for reproduction and to facilitate future research.

## 2 THINKING BACKWARD IN MCQS

### 2.1 SOLVING MCQS

**Solving MCQs with forward thinking**   As shown in Figure 1, in MCQs, LLM process a question $q$ alongside a set of answer choices $A = \{a_1, a_2, \ldots, a_n\}$. Each (question, answer) pair $(q, a_i)$ is encoded to compute a relevance score $s_i$. The model then selects the answer $a_k$ corresponding to the highest score: $k = \arg\max_i s_i$.

To compute $s_i$, the model evaluates the log-probability of generating the answer $a_i$ given the question $q$. This log-probability is often normalized to account for variations in answer length, preventing a bias toward shorter or longer responses. Various normalization techniques (Holtzman et al., 2021) can be applied, however, we resort to the most common approach which divides the total log-probability by the length of the answer $N_i = \text{len}(a_i)$ in tokens or bytes, resulting in a normalized relevance score: $s_i = \frac{\log p(a_i|q)}{N_i}$. The log-probability is factorized as

$$\log p(a_i \mid q) = \sum_{l=1}^{N_i} \log p_{L2R}(a_i^l \mid q, a_i^{<l}), \tag{1}$$

where $a_i^l$ represents the $l$-th token in $a_i$.

**Solving MCQs with reverse thinking**   With an R2L model, $s_i$ can be computed using Bayes' rule:

$$s_i = \log p(a_i \mid q)/M_i = \frac{1}{M_i}(\log p_{R2L}(q \mid a_i) + \log p_{R2L}(a_i) - C),$$

where $M_i = \text{len}(q, a_i)$, $C = \log p_{R2L}(q)$ is a constant. $\log p_{R2L}(q \mid a_i)$ and $\log p_{R2L}(a_i)$ can be autoregressively factorized in R2L manner similar to the forward thinking process in Eq. equation 1. We consider 3 paradigms of the $s_i$ for reverse thinking: (1) normalized $s_i$ with $M_i = \text{len}(q, a_i)$ resembling the forward thinking; (2) unnormalized $s_i$ with $M_i = 1$; (3) unnormalized $s_i$ without prior, i.e. $s_i = \log p_{R2L}(q \mid a_i)$.

Note that "reverse thinking" refers to both token-level reversal and the question-answer order reversal, which are coupled in our approach. During both training and inference, the R2L model processes the entire sequence (including both question and answer) in reversed token order. For example, an input template like "Question: {Q} Answer: {A}" is reversed at the token level before being fed to the R2L model during both training and evaluation.

### 2.2 MODEL EVALUATION

We conduct our evaluation on standard LLM evaluation tasks with MCQs that cover different domains including commonsense reasoning, logical reasoning, truthfulness evaluation and more.

Our evaluation tasks include HellaSwag (Zellers et al., 2019), ARC (Clark et al., 2018), MMLU (Hendrycks et al., 2021), Openbook QA (Mihaylov et al., 2018), MathQA (Amini et al., 2019), LogiQA (Liu et al., 2020), PIQA (Bisk et al., 2019), Social IQA (Sap et al., 2019), Commonsense QA (Talmor et al., 2018), Truthful QA (Lin et al., 2021), and WinoGrande (Sakaguchi et al., 2021). For ARC (easy, hard) and MMLU, we combine all the subtasks to report the overall score. We use Eleuther-AI LM-eval harness (Gao et al., 2024) for all the evaluations. For MMLU, LogiQA, and Commonsense QA, we modify the task templates to present full answer choices rather than just choice labels, following one of the standard evaluation approaches validated by reproducing FinewebEdu's official results. To verify that our findings are not template-dependent, we conducted additional experiments with reversed templates (Appendix F.1), which confirm that R2L models maintain their advantages on specific tasks regardless of template direction.

### 2.3 MODEL PRETRAINING

To pretrain the model, we first tokenize each complete dataset. The R2L model is then trained by reversing all tokens within each training data instance at the token level—that is, the entire token sequence is reversed during both pretraining and inference. The positional embeddings are also

Table 1: Comparing L2R and R2L on MCQs. All the models are trained on 350B non-repeating tokens. The HF-2B baseline is from Penedo et al. (2024). We directly used their reported numbers. EDU-2B, EDU-8B and HF-2B models are trained with the same FineWeb-EDU 350B dataset. Green indicates R2L wins, red indicates R2L loses.

| | DCLM-2B | | | EDU-2B | | | EDU-8B | | | HF-2B |
|---|---|---|---|---|---|---|---|---|---|---|
| | L2R | R2L | % Change | L2R | R2L | % Change | L2R | R2L | % Change | L2R |
| Training loss | **2.668** | 2.724 | +2.10 | **2.345** | 2.396 | +2.17 | **2.087** | 2.138 | +2.44 | - |
| **LogiQA** | 30.57 | **31.64** | +3.52 | 27.96 | **31.49** | +12.64 | 29.95 | **31.03** | +3.61 | - |
| **OpenbookQA** | 36.00 | **38.40** | +6.67 | 42.40 | **44.40** | +4.72 | 45.00 | **48.40** | +7.56 | 41.04 |
| **TruthfulQA** | 19.82 | **29.99** | +51.23 | 24.36 | **28.76** | +18.09 | 24.97 | **31.70** | +26.95 | - |
| **CommonsenseQA** | 42.83 | **45.29** | +5.74 | 42.92 | **45.13** | +5.15 | 39.15 | **44.96** | +14.84 | 36.60 |
| Social IQA | **41.56** | 40.94 | -1.48 | **42.78** | 42.22 | -1.32 | **44.58** | 43.50 | -2.42 | 40.52 |
| ARC | **54.11** | 43.88 | -18.91 | **60.65** | 52.31 | -13.75 | **68.29** | 56.22 | -17.67 | 57.47 |
| HellaSwag | **60.87** | 45.89 | -24.62 | **60.57** | 44.34 | -26.79 | **71.60** | 49.22 | -31.26 | 59.34 |
| MathQA | **26.50** | 22.21 | -16.18 | **26.80** | 24.86 | -7.25 | **28.77** | 25.33 | -11.96 | - |
| MMLU | **31.66** | 31.31 | -1.10 | **34.57** | 34.35 | -0.62 | **38.90** | 37.11 | -4.60 | 37.35 |
| PIQA | **74.43** | 58.05 | -22.00 | **74.48** | 57.13 | -23.30 | **77.80** | 59.14 | -23.98 | 76.70 |
| Winogrande | **61.01** | 53.51 | -12.29 | **60.93** | 54.85 | -9.97 | **65.75** | 54.70 | -16.81 | 57.54 |

reversed to align with the reversed sequence order, while the tokenizer itself remains unchanged from L2R (we use the Llama3 tokenizer for both). Essentially, the R2L model learns to predict the previous token given the subsequent context, using the same vocabulary as L2R but processing information in the opposite direction.

For a fair comparison between the R2L and L2R models, both models are pretrained from scratch using the same Fineweb-EDU subset dataset comprising 350B tokens (Penedo et al., 2024). Each model consists of 2B parameters (**EDU-2B**), which is the default setting in our experiments. We also train 1.5B, 4B, 8B L2R and R2L models with the same 350B Fineweb-EDU dataset, and 2B L2R and R2L models trained with a random subset of the DCLM dataset (Li et al., 2024a) containing 350B tokens (**DCLM-2B**). Both the L2R and R2L models are trained for a single epoch, ensuring each training instance is seen only once, thus the training loss should align with the validation loss. More details for model architecture and training are provided in Appendix B.

## 2.4 RESULTS

We present our results in Table 1. To verify our pretraining pipeline, we first compare the performance of our pretrained model with the 2B model trained by Huggingface (Penedo et al., 2024) (**HF-2B**) [1]. Under similar model size and the same dataset, our 2B model (**EDU-2B**) achieves performance comparable to or exceeding the L2R results reported by Huggingface **HF-2B**. Full results for all models ranging from 1.5B to 8B parameters are provided in our appendix C.

We compared L2R and R2L model performance across all evaluated tasks, employing bootstrap sampling (5 replicates, each with 80% resampling with replacement) for statistical robustness. As shown in Table 1, R2L models with reverse thinking exhibited significantly better reasoning performance on 4 out of 11 tasks: LogiQA, OpenBookQA, TruthfulQA, and CommonsenseQA. Statistical significance results are presented in Table 8.

These results remained consistent across different model sizes (1.5B to 8B), datasets (DCLM, FineWeb EDU), and random seeds, indicating the findings are not due to random fluctuation. The relative performance gain or loss when switching to R2L remained generally stable as model size increased (see appendix C).

For TruthfulQA specifically, we observed the most significant performance gain with R2L the improvement was substantial (51.23% on DCLM-2B). We hypothesize that the "reverse thinking" may inherently align better with truthfulness assessment, as it evaluates the question based on each answer choice rather than generating answers from the question. This framing might help the R2L model better discern subtle inaccuracies that an L2R model might overlook due to "surface form competition". Additionally, R2L models demonstrate significantly lower conditional entropy on

---

[1] https://huggingface.co/spaces/HuggingFaceFW/blogpost-fineweb-v1

TruthfulQA compared to L2R models, which aligns with our hypothesis that lower conditional entropy is associated with higher task accuracy.

For reverse thinking with R2L, we use the paradigm 3 (*i.e.*, unnormalized $s_i$ without prior) for downstream tasks evaluation. We compare the three paradigms for reverse thinking in Appendix D, Table 5. Ideally, $s_i$ should incorporate priors, as in paradigm 1 or 2. However, in practice, using $s_i$ without prior (paradigm 3) consistently yields the best performance except for Social IQA and PIQA. We hypothesize this may be due to intrinsic difficulty of estimating the prior probabilities $p(a)$ using LLMs, due to the "surface competition" calibration issues (Holtzman et al., 2021). We provide detailed explanation of our hypothesize using an illustrative example in Appendix F.

The improvement is not solely attributable to the scoring formulation, but to the synergy between R2L factorization and this unnormalized scoring. Paradigm 3 is a key calibration strategy that allows the R2L model to realize its potential by enforcing a uniform prior, which effectively alleviates the "surface form competition" that plagues L2R models. This scoring paradigm is unique to R2L factorization—it naturally provides length normalization since all choices predict the same fixed-length question. To verify that gains come from factorization itself and not just scoring, we conducted reversed template experiments (Appendix F.1), which demonstrate that R2L models trained with reversed factorization outperform L2R models even when both use similar scoring approaches.

Intuitively, paradigm 3 which uses $p(q \mid a)$ is sensible. In MCQs, answer choices are typically well-formed and reasonable text, meaning their prior probabilities $p(a)$ are unlikely to vary significantly among choices, assigning a uniform prior is probably a reasonable approach. Consequently, $p(a \mid q)$ and $p(q \mid a)$ tend to be highly correlated. Consider a real-world example from Openbook QA: for the question ($q$) "A magnet will stick to", candidate answers ($a_i$) include "a belt buckle", "a wooden table", "a plastic cup", and "a paper plate". A model can deduce that "a belt buckle" is far more likely to be associated with the question "A magnet will stick to" compared to the other options, demonstrating how $p(q \mid a)$ can effectively capture the relevance between question and answer.

We also monitor the training loss for pretraining the models on both directions. We observed findings similar to Papadopoulos et al. (2024) in that L2R yields a lower loss compared to R2L, even though both model the same target data distribution. In Papadopoulos et al. (2024), the largest model that was trained had 405M parameters while our models were trained at the popular small LLM size range of 2B-8B parameters. At this size, we observe a similar percentage difference as reported by previous work, of about 2%-2.5% increase in loss when using R2L, indicating learning the R2L factorization is more challenging. This makes it particularly interesting that on a bunch of MCQ tasks we see the R2L is performing better, as elaborated above.

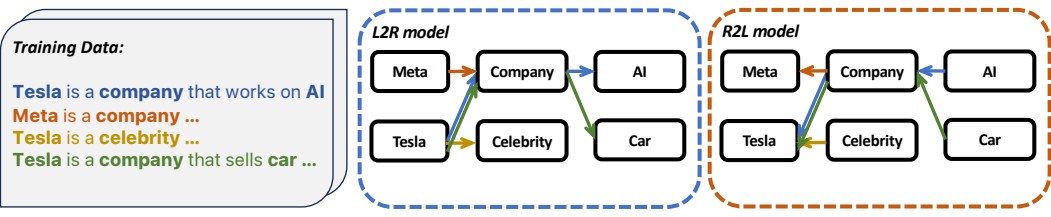

Figure 2: L2R and R2L LLMs pretrained on the same data will generate opposite search graphs based on the order in which they process the information entities.

## 3 WHAT MAKES THE PREFERRED ORDER OF THINKING?

We then seek to gain a deeper understanding of why there is a preferred orientation for the MCQs. We explore three main hypotheses (3**C**): *Calibration*, *Computability*, and *Conditional entropy*. Admittedly, there may be other factors that we have overlooked that contributes to this preference.

### 3.1 CALIBRATION

The first potential explanation concerns the scoring mechanism in forward thinking, where $s_i = \log p_{L2R}(a_i \mid q)$. Eq. equation 1 might not lead to an optimal estimation of $p(a|q)$ as it suffers

from several calibration issues. Among the choices, some may contain more words that are highly predictable (e.g., "Hong Kong" or stop-words like "a"), potentially leading to spuriously inflated relevance scores. Additional, Holtzman et al. (2021) shows that simple probability normalization in MCQs is challenging because different surface forms of semantically equivalent answers compete for probability mass, potentially *diluting* scores for correct answers due to this "surface form competition".

In contrast, reverse thinking with paradigm 3, where $s_i = \log p_{R2L}(q \mid a_i)$, mitigates this issue since the target question $q$ remains constant across all choices. We provide rationale analysis on how R2L paradigm 3 alleviates "surface competition" in Appendix F. In a nutshell, forward thinking suffers from surface form competition, where semantically similar words (e.g., "dog" and "puppy") split probability mass, reducing the likelihood of selecting the correct answer. Reverse thinking mitigates this by enforcing a uniform prior, eliminating competition in the prior distribution and allowing a fairer comparison between answer choices. This suggests that reverse thinking inherently "auto-normalizes" different choices, resulting in more robust evaluation. However, this sole theory fails to explain why reverse thinking does not consistently outperform forward thinking across all tasks, instead showing superior performance only in specific MCQ scenarios.

## 3.2 COMPUTABILITY

A second potential theoretical explanation, which echoes with Papadopoulos et al. (2024), suggests that computational complexity may underlie these directional preferences. Drawing an analogy to number theory, where multiplying prime numbers is computationally straightforward, while the reverse operation of prime factorization is NP-hard.

It is tempting to consider this computational complexity asymmetry as the main underlying cause for why L2R or R2L is preferred for specific tasks. However, recent research (Mirzadeh et al., 2024; Kambhampati, 2024; Valmeekam et al., 2024) find that LLMs may not actually perform genuine reasoning or computing, as evidenced by their poor generalization when tasks undergo minor modifications. This implies that LLMs mainly emulate *reasoning patterns* from their training data instead of carrying out actual logical computation, weakening the hypothesis that directional preferences stem from varying computability in different directions. Furthermore, most MCQs primarily involve knowledge retrieval and basic reasoning, which might not reach the complexity threshold where computational hardness would become a significant factor. Therefore, acknowledging that computability may be a factor, we keep exploring alternative hypotheses.

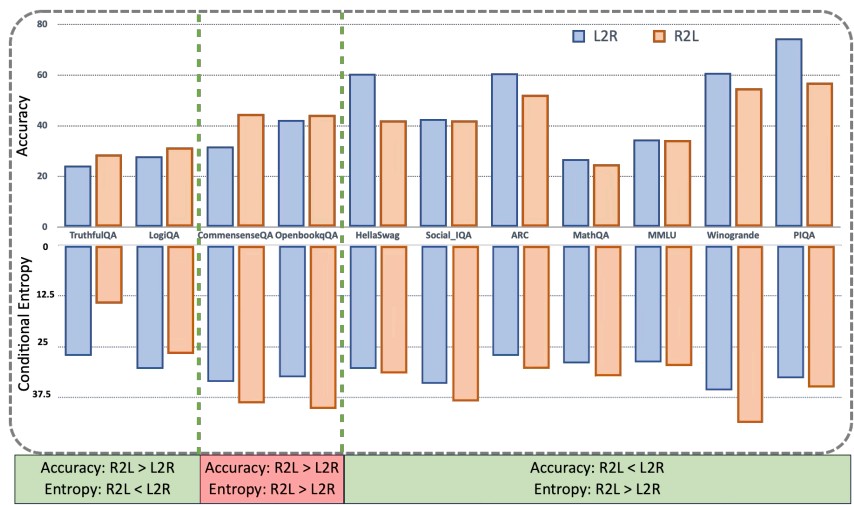

Figure 3: Lower conditional entropy may associate with higher accuracy in the reasoning direction.

## 3.3 CONDITIONAL ENTROPY

Our final hypothesis posits that the optimal direction of thinking is closely related to the *conditional entropy* of the downstream task. Recent work has shown that learning knowledge extraction and simple multihop reasoning is more challenging for problems with higher degree of **branching factors** or "**globality degree**" compared to those with lower branching factors and more deterministic relationships (Abbe et al., 2024). It is conceivable that directionality of data can impact the branching degree and lead to different learning efficiencies in different directions (for example multiplication in left-to-right direction is factorization in the opposite direction, each with different branching factors).

Previous work (Berglund et al., 2023; Allen-Zhu & Li, 2023b) has also demonstrated that LLMs suffer from the "reversal curse", indicating that inverse R2L search in LLMs is inherently challenging for L2R models - due the disconnect between training and inference directions. Consider an LLM trained on sequences of knowledge/information name entities $(e_1, e_2, \cdots, e_n)$. LLM may effectively construct a **directed** search graph that maps the key $(e_1, \cdots, e_{i-1})$ to the value $e_i$ for any $i$. Following this logic, the training data essentially forms a Bayesian network that can be represented as a *directed acyclic graph* (DAG) of entities. Similarly, training an R2L model yields an analogous DAG but with reversed edge directions (see Figure 2 for an illustration). The search efficiency between these two graphs may vary given different queries.

We hypothesize here that between two different factorizations of the data, **the direction yielding lower conditional entropy will perform better in MCQs**, as it reflects better efficiency in knowledge extraction and multi-hop search. We note however, that this is only true when models under both factorization directions have sufficiently low error, which seems to be true for our models here.

More formally, for a downstream MCQ task $T$ with question and answer choices following task-specific data distribution $P_T(q, a)$, we compare the *conditional entropy* in both directions under pretrained L2R and R2L models (Eq. equation 2 for L2R and Eq. equation 3 for R2L):

$$- \mathbb{E}_{q' \sim P_T(q)} \sum_a p_{L2R}(a|q') \log p_{L2R}(a|q'). \tag{2}$$

$$- \mathbb{E}_{a' \sim P_T(a)} \sum_q p_{R2L}(q|a') \log p_{R2L}(q|a'). \tag{3}$$

We assume that the conditional entropy is a proxy for the quality of the learned model, and the direction with lower conditional entropy should perform better. However, computing these summations in equation 2 and equation 3 is intractable due to the exponentially large candidate space. Therefore, we employ Monte Carlo estimation of equation 2 and equation 3 as proxy measures, specifically computing

$$- \mathbb{E}_{q' \sim P_t(q), a' \sim p_{L2R}(a|q')} \log p_{L2R}(a'|q'), \tag{4}$$
$$- \mathbb{E}_{a' \sim P_t(a), q' \sim p_{R2L}(q|a')} \log p_{R2L}(q'|a'). \tag{5}$$

Because of the extensive amount of evaluation datasets, due to limited computation budget, we only conducted a single sample rollout for $a' \sim p_{L2R}(a|q')$ and $q' \sim p_{R2L}(q|a')$. We recognize that this may not be a precise representation of the true conditional entropy, given that the candidate space grows exponentially with the maximum sequence length.

**Empirical Verification** To verify this hypothesis, we estimate the conditional entropy for all the evaluation tasks. We provide more experimental details in Appendix E. Figure 3 presents our empirical results using single-sample Monte Carlo estimation. While this initial analysis shows a general trend supporting our hypothesis, we acknowledge that single-sample estimation is high-variance and may not be fully reliable. To address this limitation, we conducted additional experiments with 10 Monte Carlo samples per task. The improved estimates, reported with standard errors in Appendix Table 6, demonstrate that **9 out of 11 benchmarks follow the trend where lower conditional entropy correlates with higher accuracy**. For example, TruthfulQA shows L2R conditional entropy of $26.21 \pm 3.82$ nats versus R2L's $15.25 \pm 2.45$ nats, and LogiQA shows $30.57 \pm 0.91$ (L2R) versus $27.53 \pm 0.82$ (R2L), both cases where R2L outperforms L2R in accuracy.

Two tasks (CommonsenseQA and OpenbookQA) show exceptions to this pattern. We hypothesize that for these specific tasks, the *Computability* factor dominates the conditional entropy effect. These

tasks are highly reliant on commonsense and factual knowledge retrieval where both the question (Q) and answer (A) are often short phrases. For $p(q|a)$, the model reconstructs or validates a short question from a short answer—a compact information loop. For $p(a|q)$, the model generates a short answer from a short question. The complexity may stem not from the length of dependency (which CE measures) but from the density of required knowledge and inherent ambiguity of the text distribution, making the computational process itself the primary performance bottleneck regardless of factorization direction.

Importantly, the conditional entropy principle is validated in our controlled arithmetic simulation (Section 4), where confounding factors are isolated and lower conditional entropy consistently predicts better performance. The real-world task results therefore serve to illustrate that the CE principle exists but can be overridden by other factors (like high Computability bottlenecks or knowledge density) in complex linguistic domains. In Figure 3, we observed that the conditional entropy of R2L is generally greater than L2R. This trend could be related to the findings presented in Table 1, indicating that R2L tends to have higher training loss too. Complementing the rationale in Papadopoulos et al. (2024), we hypothesize that the ease with which the language model can approximate the factorized distribution of L2R and R2L, may be also tied to which direction exhibits higher branching factors in that direction. We leave this exploration for future study.

Table 2: Results of the controlled simulation study of 4-digits multiplication. Theoretical Conditional Entropy (Theo. Cond. Ent.) represents the expected conditional entropy under an ideal model. L2R consistently outperforms R2L in Forward X, while R2L is superior in Reverse X. Lower conditional entropy correlates with higher accuracy.

| | Forward X | | | Reverse X | | |
|---|---|---|---|---|---|---|
| | L2R | R2L(m,n) | R2L(m) | R2L | L2R(m,n) | L2R(n) |
| Test Accuracy (%) | **99.81**±0.15 | 59.71±1.99 | 60.93 ± 0.88 | **100**±0 | 97.82±0.35 | 99.85±0.10 |
| Train Accuracy (%) | **99.76**±0.15 | 59.03 ± 1.66 | 61.22±1.12 | **100**±0 | 97.90±0.42 | 99.98±0.04 |
| Test Cond. Ent. (nats) | 0.06 | 1.18 | 0.08 | 0 | 0.84 | 0.01 |
| Train Cond. Ent. (nats) | 0.06 | 1.17 | 0.08 | 0 | 0.83 | 0.01 |
| Theo. Cond. Ent. (nats) | 0 | 1.49 | 0 | 0 | 1.49 | 0 |
| Training loss | **0.86** | 0.94 | 0.94 | **0.86** | 0.94 | 0.94 |

## 4 CONTROLLED SIMULATION STUDY

The three hypotheses discussed in Section 3 are intricately entwined in actual MCQs, making it challenging to disentangle them. To better investigate the hypotheses explaining the optimal direction for MCQs, we conducted a meticulously controlled simulation study (Figure 4) focused on 4-digit multiplication. Although the arithmetic dataset is different from the real language modeling datasets, this simulation study can be a good controlled experiment to understand the phenomenon we observed in Section 3, as we are investigating the underlying principle of the model with different factorizations regardless of the dataset. The L2R and R2L models were initialized **from scratch** and **exclusively** trained on this simulation dataset to eliminate any potential confounding factors. All data instances share the same format and length, removing the *calibration* effect from the analysis and allowing us to concentrate on *computability* and *conditional entropy*.

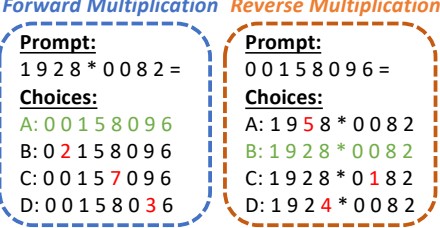

Figure 4: Simulation Study. Forward multiplication simulates a **many-to-one** mapping scenario, while reverse multiplication simulates a **one-to-many** mapping.

**Experiment Setup** We conduct two types of simulation experiments: Forward Multiplication (**Forward X**) and Reverse Multiplication (**Reverse X**). In Forward X, each training instance was represented as $m \times n = p$, where $m, n \in \{0, \ldots, 10^4\}$ and $p \in \{0, \ldots, 10^8\}$. The formatting included spaces between digits and mathematical operators to ensure a consistent single tokenization for both L2R and R2L models. In Reverse X, the multiplication was in reverse order, such as

$p = m \times n$. For each simulation type, L2R and R2L models were trained with a 2B model size with 1 epoch on all $10^8$ non-repeating equations except 1,000 test examples, totaling to almost 3.2B tokens.

The model performance was assessed using the held-out test set with 1,000 examples. These examples were converted into a multiple-choice format consisting of 4 choices (Figure 4). Other than the correct answer, the remaining three hard-negative options were created by altering a single digit in the correct answer to a random other digit, at a random position. The presenting order of the four choices are then randomly shuffled. We augmented the test set 10 times to calculate the average metrics.

As multiple pairs of $m$ and $n$ can be mapped to the same product $p$, Forward X is a **many-to-one** mapping. The theoretical conditional entropy for predicting the correct $p$ from $m \times n$ is 0 under an oracle model. However, as there are several paths from the product $p$ to the $m, n$ pairs, the theoretical conditional entropy for predicting the $m \times n$ from $p$ becomes 1.49 nats under an oracle model. In the Reverse X task, which transitions into a **one-to-many** scenario, the analysis is inverted.

For Forward X, we explore an alternative R2L evaluation method, denoted as **R2L(m)**, where the relevance score of the i-th choice $p_i$ is calculated as $s_i = \log p_{R2L}(m \mid p_i, n)$, focusing on the conditional entropy of $m$ rather than $m \times n$ as in the standard **R2L(m,n)** method. Since R2L(m) is essentially division, it is deterministic with a theoretical conditional entropy of 0. Similarly, we have a variant for L2R in reverse X, called **L2R(n)**.

**Results** The results are presented in Table 2. In Forward X scenarios, L2R models demonstrate higher accuracies than R2L(m,n) models, with correspondingly lower conditional entropy and training loss. This observation aligns with our hypothesis in Section 3. Conversely, in Reverse X scenarios, the R2L model outperforms the L2R(m,n) model. The training and test performance gaps are minimal.

Interestingly, R2L(m) achieves better accuracy than R2L(m,n) in Forward X as conditional entropy decreases. Similarly, L2R(m,n) surpasses L2R(n) in Reverse X. This suggests that **when maintaining the same thinking direction – where computability should remain equivalent – performance improvements can be achieved** by configuring $s_i$ to have lower conditional entropy. This hints that the R2L performance on MCQs can potentially be further improved by configuring the input to predict fewer tokens in the question $q$, so that the minimum conditional entropy is obtained. We leave this for future exploration.

On the other hand, comparing L2R with R2L(m), where theoretical conditional entropy equal 0, L2R maintains superiority, indicating that **computability likely remains as a key factor**. For the Reverse X task, the accuracy gap between R2L and L2R(n) is smaller than the accuracy gap between L2R(m,n) and L2R(n), suggesting that the conditional entropy may explain more of the performance gap than the computability. Notably, models achieve higher accuracies on Reverse X compared to their Forward X counterparts, despite similar training loss and conditional entropy values. This disparity could probably be attributed to the closer proximity of choices in Forward X, which inherently increases task difficulty. We provide additional analysis comparing Forward X and Reverse X in Appendix G.

## 5 RELATED WORK

**Reversal Curse** Berglund et al. (2023) first investigates the "reversal curse" in LLMs, which refers to the phenomenon where models trained on forward text data struggle to perform well on inverse search tasks. Allen-Zhu & Li (2023a) further discusses this issue and proposes that augmentation during the pretraining stage can help bridge the knowledge extraction performance gap in reverse entity mapping. In a similar vein, Golovneva et al. (2024) suggest training a unified model that combines text data with augmented reversed or partially reversed data can mitigate the reversal curse. These studies imply that autoregressively-trained language models tend to have a linear and unidirectional thinking process, and certain types of augmentation can facilitate the model in making complex connections between pieces of learned information to enable more intricate cross-referencing. Our research also demonstrates that the autoregressive nature of LLMs may introduce inductive biases rooted from the pretraining corpus. Instead of focusing on the "reversal curse," we suggest that knowledge extraction and reasoning may be more straightforward in the direction with lower conditional entropy.

**Order of Reasoning** Previous works have also been exploring the reasoning order's impact to the reasoning performance. Vinyals et al. (2015) first demonstrates that the sequence in which input and output data are organized significantly impacts the performance of sequence-to-sequence models and

propose to search over possible orders during training to manage unstructured output sets. Recently, Papadopoulos et al. (2024) reveals a surprisingly consistent lower log-perplexity when predicting in L2R versus R2L, despite theoretical expectations of symmetry. The authors attributes this asymmetry to factors like sparsity and computational complexity. We also observe this difference yet we have another hypothesis rationale beyond theirs. Zhang-Li et al. (2024) shows that by reversing the digit order, prioritizing the least significant digit can improve LLMs's performance on arithmetic, which aligns with our findings in Section 4.

Previous studies on sequence modeling have also delved into relaxing the conventional "left-to-right" autoregressive dependencies, primarily to facilitate parallel generation (Gu et al., 2018; Ghazvininejad et al., 2019; Gu & Kong, 2021; Zhang et al., 2020) and non-monotonic generation (Welleck et al., 2019; Gu et al., 2019). Text diffusion has recently emerged as a promising approach in terms of planning and controllability (Li et al., 2022; Zhang et al., 2023; Gong et al., 2024). It has shown to be more effective than LLM than language model (LLM), particularly for tasks that require bidirectional reasoning strategies such as sudoku and countdown games (Ye et al., 2024). Alternatively, the Belief State Transformer (BST) (Hu et al., 2025) enhances sequence modeling by using both prefix and suffix inputs to predict subsequent and preceding tokens, effectively capturing a compact belief state for improved goal-conditioned decoding and test-time inference. In contrast, our work primarily investigates the optimal reasoning order for non-generative tasks requiring structured inference.

**Multiple-Choice Questions (MCQs) for LLM evaluation** MCQs have been widely used for evaluating LLM's reasoning and knowledge extraction abilities. Zheng et al. (2023) demonstrates that LLMs exhibit a selection bias in MCQs, favoring certain option positions, and introduces a debiasing method to mitigate this issue. Pezeshkpour & Hruschka (2023) examines how LLMs' performance on MCQs is influenced by the order of answer options, finding that reordering can lead to huge performance variations. Ghosal et al. (2022) proposes reframing MCQs as a series of binary classifications, demonstrating that this approach significantly improves performance across various models and datasets. Li et al. (2024b) highlights issues like positional biases and discrepancies compared to long-form generated responses, when using MCQs in evaluating LLMs. Wiegreffe et al. (2024) discovers that the prediction of specific answer symbols is primarily attributed to a single middle layer's multi-head self-attention mechanism, with subsequent layers increasing the probability of the chosen answer in the model's vocabulary space. In contrast to the previous work, our work first shows the connection between the preferred reasoning direction and the direction that has lower conditional entropy in MCQ evaluations.

## 6 CONCLUSION

In this work, we investigated what makes the preferred thinking direction for LLMs in multiple-choice questions. Through extensive experimentation with models of varying sizes and training datasets, we discovered the surprising finding that R2L factorization can outperform traditional L2R approaches in specific MCQ tasks (4 out of 11 evaluated benchmarks). This finding challenges the prevailing assumption that L2R is universally optimal and demonstrates that the preferred factorization direction is task-dependent. Our analysis revealed that the effectiveness of each factorization direction may be intrinsically linked to several factors including calibration, computability, and conditional entropy of the downstream task distribution, with lower conditional entropy generally yielding better performance. We disentangle and validate these factors through controlled simulation studies using arithmetic tasks.

The core contribution of this work lies in bringing this phenomenon to the community's attention and initiating analysis into the underlying factors that might explain why a particular thinking direction is preferred in specific task domains. We emphasize that our findings are domain-specific rather than universal—R2L is not a general replacement for L2R, but demonstrates that alternative factorizations can be advantageous when the task distribution exhibits certain statistical properties. These findings may suggest the potential for future language model development by revealing the knowledge extraction and reasoning machinery of LLMs and suggesting that alternative or hybrid factorizations deserve serious consideration in model design. We also discussed the limitation of this work in Appendix A. Future work could explore additional factorization strategies beyond L2R and R2L, investigate applications to other types of language tasks, and develop more sophisticated methods for combining different factorizations based on task characteristics.

**Reproducibility Statement:** We document evaluation metrics, ablations, model architectures, datasets, preprocessing, and training protocols in Appendix, section 2.2, section 2.3 and section 4; Theoretical details (step-aware conditional entropy analysis, controlled simulation methodology, factorization comparison framework) appear in section 3 and section 4. The code and model checkpoints are publicly accessible.

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
