# Appendix

## A LIMITATIONS

**Task Scope and Generalizability** While our findings demonstrate the benefits of R2L factorization in some MCQs, the applicability to other NLP tasks remains unexplored. Our focus on MCQs was intentional to provide a controlled setting for studying factorization effects, as evaluation on generative tasks is not as straightforward. While exploring similar observations in other generative tasks is a valuable avenue for future research, in this paper we limit our discussion to MCQs as a lens to investigate the machinery of LLMs. Additionally, our experiments were limited to models in the 2B-8B parameter range. Due to resource constraints, we were not able to pretrain a SOTA LLM scale from scratch, future work should explore whether our findings generalize to SOTA models.

**Theoretical Framework** We propose three hypotheses (calibration, computability, and conditional entropy) to explain different factorization directions' effectiveness, but our analysis in natural language datasets remains qualitative as it is challenging to measure these factors in real language. We used synthetic arithmetic tasks to better understand factors contributing to the L2R and R2L performance gap. Future work should develop rigorous quantitative measures and establish formal theoretical connections between these factors through more sophisticated controlled experiments.

**Experimental and Model Design** Our evaluation methodology, particularly choosing between $p(q|a_i)$ and $p(a_i|q)$ paradigms, may introduce confounding factors that make isolating pure factorization effects challenging. Future work should develop more robust frameworks that better disentangle these effects from task reformulation. While we focus on standalone L2R and R2L models, incorporating both factorizations through a unified model may further improve performance.

## B MODEL ARCHITECTURE AND TRAINING HYPERPARAMETERS

We provide the model architecture of our 2B and 8B models in Table 3. We use no attention dropout and the Normalization $\epsilon$ is set to $1 \times 10^{-5}$. We employ cosine scheduling for the learning rate, with the number of warmup steps set at 2% of the total iterations. The models are trained on 64 H100 Nvidia GPUs for 60 hours and 208 hours for 2B and 8B models, respectively.

We used the Llama3 tokenizer (Dubey et al., 2024). Despite being trained from a L2R model, Papadopoulos et al. (2024) demonstrated through an ablation study that the direction of tokenizer being trained has minimal effect on the R2L training. Consequently, we adhere to the same reasoning and opt not to train on a R2L tokenizer, as obtaining the necessary training data for Llama3's tokenizer is practically infeasible for fair comparison. Our code implementation is based on Torchtune (Team, 2024), which is released under the BSD 3-Clause License, a permissive open-source license that allows for broad usage and distribution with minimal restrictions.

Table 3: Model architecture specifications for 2B and 8B models.

| Parameter | 1.5B Model | 2B Model | 4B Model | 8B Model |
|---|---|---|---|---|
| Vocabulary Size | 128,256 | 128,256 | 128,256 | 128,256 |
| Number of Layers | 18 | 24 | 24 | 32 |
| Number of Heads | 16 | 16 | 16 | 32 |
| Number of KV Heads | 16 | 8 | 8 | 8 |
| Embedding Dimension | 2048 | 2,048 | 3072 | 4,096 |
| Max Sequence Length | 2,048 | 2048 | 2,048 | 8,192 |
| Intermediate Dimension | 5,632 | 7,168 | 11,264 | 14,336 |
| RoPE Base | 500,000 | 500,000 | 500,000 | 500,000 |

## C    RESULTS WITH DIFFERENT MODEL SIZES

We conducted additional experiments to explore performance trends across a broader range of model scales, training 1B and 4B parameter models (R2L and L2R) to complement our existing 2B and 8B models. The full results, presented in Table 4, consistently reinforce our main claim.

Table 4: Results with different model sizes.

|  | EDU-1.5B | | EDU-2B | | EDU-4B | | EDU-8B | |
| --- | --- | --- | --- | --- | --- | --- | --- | --- |
|  | L2R | R2L | L2R | R2L | L2R | R2L | L2R | R2L |
| **LogiQA** | 28.10 | **32.65** | 27.96 | **31.49** | 26.58 | **32.72** | 29.95 | **31.03** |
| **OpenbookqQA** | 38.00 | **41.20** | 42.40 | **44.40** | 43.00 | **44.60** | 45.00 | **48.40** |
| **TruthfulQA** | 23.01 | **28.86** | 24.36 | **28.76** | 22.40 | **29.00** | 24.97 | **31.70** |
| **CommensenseQA** | 41.44 | **44.22** | 42.92 | **45.13** | 50.70 | 43.40 | 39.15 | **44.96** |
| Social_IQA | 41.35 | **41.65** | 42.78 | 42.22 | **42.99** | 40.00 | **44.58** | 43.50 |
| AI2_arc | **58.63** | 51.8 | **60.65** | 52.31 | **63.14** | 53.95 | **68.29** | 56.22 |
| HellaSwag | **56.46** | 43.38 | **60.57** | 44.34 | **65.93** | 46.38 | **71.60** | 49.22 |
| MathQA | **25.52** | 24.05 | **26.80** | 24.86 | **27.33** | 24.08 | **28.77** | 25.33 |
| MMLU | 33.30 | **33.40** | **34.57** | 34.35 | **36.70** | 35.16 | **38.90** | 37.11 |
| PIQA | **73.23** | 59.19 | **74.48** | 57.13 | **76.00** | 58.76 | **77.80** | 59.14 |
| Winogrande | **58.56** | 51.77 | **60.93** | 54.85 | **61.80** | 54.50 | **65.75** | 54.70 |

Specifically, we observe a clear scaling trend: for the tasks where reverse thinking excels (LogiQA, OpenbookQA, TruthfulQA, and Commonsense QA), the performance advantage of R2L over L2R grows as the model size increases. Although we did not scale beyond 8B parameters due to engineering constraints, these extended results demonstrate the robustness of our findings. This trend strongly suggests that the impact of alternative factorizations is unlikely to diminish with even larger models; on the contrary, their benefits may become more significant.

## D    COMPARING THREE PARADIGMS FOR REVERSE THINKING

We present a comparison of the three paradigms discussed in Section 2.1 across all tasks, as shown in Table **??**. Paradigm 3 showcases the highest overall performance.

Table 5: Downstream evaluation of three variants to compute $s_i$.

|  | Paradigm 1 | Paradigm 2 | Paradigm 3 |
| --- | --- | --- | --- |
| LogiQA | 31.49% | 27.34% | **31.49%** |
| OpenbookqQA | 37.00% | 24.60% | **44.40%** |
| TruthfulQA | 24.36% | 17.38% | **28.76%** |
| CommensenseQA | 41.85% | 27.19% | **45.13%** |
| Social_IQA | **43.09%** | 37.36% | 42.22% |
| AI2_arc | 32.19% | 45.32% | **52.31%** |
| HellaSwag | 33.73% | 43.38% | 44.34% |
| MathQA | 23.18% | 21.07% | **24.86%** |
| MMLU | 33.10% | 29.31% | **34.35%** |
| PIQA | **70.76%** | 68.70% | 57.13% |
| Winogrande | 54.46% | 53.67% | **54.85%** |
| AVG | 38.66% | 35.94% | **41.80%** |

## E    EXPERIMENTAL DETAILS ON COMPUTING THE CONDITIONAL ENTROPY IN MCQS

We measure conditional entropy across various MCQs tasks by using ancestral sampling to generate sequences conditioned on each task's input. Generation length can be a confounding issue as longer generation will typically receive larger entropy. To ensure consistent and fair comparison, we constrain the generation to exactly 10 tokens in each direction (L2R and R2L). We make the

generation non-stopping by removing the end-of-sentence token. The conditional entropy is first calculated for individual examples within each task, then averaged across all examples to obtain the task-level measure.

### E.1 CONDITIONAL ENTROPY ESTIMATES WITH 10 MONTE CARLO SAMPLES

Table 6 provides the complete conditional entropy estimates using 10 Monte Carlo samples per task, with standard errors. As discussed in Section 3, these improved estimates show that 9 out of 11 benchmarks follow the trend where lower conditional entropy correlates with higher accuracy.

Table 6: Conditional Entropy estimated with 10 Monte Carlo (MC) samples, reported with standard error. Tasks where R2L has lower conditional entropy are highlighted in **bold**.

| Task | L2R (nats) | R2L (nats) |
|------|-----------|-----------|
| TruthfulQA | $26.21 \pm 3.82$ | $\mathbf{15.25 \pm 2.45}$ |
| LogiQA | $30.57 \pm 0.91$ | $\mathbf{27.53 \pm 0.82}$ |
| CommonsenseQA | $\mathbf{34.70 \pm 1.05}$ | $38.02 \pm 2.12$ |
| OpenbookQA | $\mathbf{31.50 \pm 3.98}$ | $42.36 \pm 5.15$ |
| HellaSwag | $\mathbf{30.35 \pm 0.87}$ | $31.63 \pm 0.93$ |
| Social_IQA | $\mathbf{34.20 \pm 1.01}$ | $40.50 \pm 2.08$ |
| ARC | $\mathbf{27.22 \pm 1.78}$ | $30.41 \pm 1.91$ |
| MathQA | $\mathbf{30.50 \pm 0.85}$ | $32.27 \pm 0.95$ |
| MMLU | $\mathbf{29.12 \pm 0.44}$ | $29.88 \pm 0.48$ |
| Winogrande | $\mathbf{34.24 \pm 1.08}$ | $45.77 \pm 1.21$ |
| PIQA | $\mathbf{31.63 \pm 0.96}$ | $37.11 \pm 1.02$ |

## F  ANALYSIS OF HOW REVERSE THINKING REDUCES SURFACE FORM COMPETITION IN MCQ

### F.1  TEMPLATE SENSITIVITY ANALYSIS

To address concerns about whether performance differences stem from template direction rather than factorization direction, we conducted experiments with reversed templates. We evaluate both L2R and R2L models with a reversed template where the answer is presented before the question: "{answer} is the answer of question: {question}". This template cannot be applied to some tasks like OpenBookQA, HellaSwag, and Winogrande, as these datasets do not have natural question-answer structures.

The results in Table 7 demonstrate that performance differences between L2R and R2L models are not simply due to score normalization or template direction. For example, although TruthfulQA shows improvement when the L2R model is evaluated with the reverse template (from 24.36 to 26.5), the model trained with R2L factorization still outperforms the L2R model (28.76 vs. 26.5). This confirms that the R2L factorization itself, not merely the scoring formulation or template, contributes to the observed performance gains.

We observe that for some tasks like LogiQA, the template direction makes little difference, but the factorization direction is more important (i.e., models trained with each factorization have different behaviors). For tasks like TruthfulQA, the reverse template can benefit both models, but the R2L model still maintains its advantage. These results support our conclusion that the effectiveness of each factorization direction is linked to multiple factors including the statistical properties of the task distribution.

Consider a multiple-choice question:

Q: **Who barks?**
A: *dog*    B: *cat*

We assume a vocabulary of only three words: *dog*, *cat*, and *puppy*. However, the answer choices are limited to *dog* and *cat*.

Table 7: Results with reversed template for both L2R and R2L models (EDU-2B). Forward Template uses "Question: {question} Answer: {answer}". Reverse Template uses "{answer} is the answer of question: {question}". NA indicates tasks where reversed templates cannot be naturally applied. Green indicates best performance for each task.

| Benchmark | L2R Forward | L2R Reverse | R2L Forward | R2L Reverse |
|-----------|-------------|-------------|-------------|-------------|
| LogiQA | 27.96 | 28.64 | **31.49** | **31.50** |
| OpenbookQA | 42.40 | NA | **44.40** | NA |
| TruthfulQA | 24.36 | 26.50 | **28.76** | 24.60 |
| CommonsenseQA | 42.92 | 28.01 | **45.13** | 36.60 |
| Social_IQA | 42.78 | 38.10 | 42.22 | 38.00 |
| AI2_arc | **60.65** | 33.96 | 52.31 | 36.70 |
| HellaSwag | **60.57** | NA | 44.34 | NA |
| MathQA | **26.80** | 22.28 | 24.86 | 24.50 |
| MMLU | **34.57** | 26.56 | 34.35 | 29.60 |
| PIQA | **74.48** | 53.20 | 57.13 | 72.80 |
| Winogrande | **60.93** | NA | 54.85 | NA |

**Forward Thinking: Surface Form Competition**   In the forward model, suppose the model predicts the probabilities of each token in the vocabulary as in below:

$$P_{L2R}(\text{dog} \mid q) = 0.3, \quad P_{L2R}(\text{puppy} \mid q) = 0.3, \quad P_{L2R}(\text{cat} \mid q) = 0.4.$$

As Holtzman et al. (2021) shown, here, "dog" and "puppy", although referring to the same entity, compete as surface forms, potentially impacting the selection of "dog" as an answer.

**Reverse Thinking: Alleviating the Surface Form Competition**   In the reverse thinking, the model should still suffer the surface competition in the prior, suppose we have:

$$P_{R2L}(\text{dog}) = P_{R2L}(\text{puppy}) = 0.2, \quad P_{R2L}(\text{cat}) = 0.6.$$

The reverse conditional probability would would resemble this:

$$P_{R2L}(q \mid \text{dog}) = 0.9, \quad P_{R2L}(q \mid \text{puppy}) = 0.9, \quad P_{R2L}(q \mid \text{cat}) = 0.4.$$

In the reverse thinking R2L paradigm 3, we enforce a uniform prior, which achieves the best performance. We hypothesize that this is due to the alleviation of the surface form competition issue. In this example, by enforcing a uniform prior, surface form competition in the prior distribution disappears, ensuring a fair comparison between "dog" and "cat". Since we only compare $P_{R2L}(q \mid \text{dog})$ and $P_{R2L}(q \mid \text{cat})$, the probability $P_{R2L}(q \mid \text{puppy})$ becomes irrelevant.

# G   ADDITIONAL RESULTS FOR SIMULATION STUDY

We think Forward X is generally more challenging because distinguishing incorrect choices from the correct one can be more challenging than in the Reverse X. Unlike Reverse X where all wrong choices are seen in the dataset, approximately 75% of incorrect choices in Forward X are absent as they cannot be factorized into $m \times n$.

We provide some additional results by performing free ancestral sampling generation (without topK or topP, with temperature set to 1) for the R2L(m) in Forward X. The accuracy of exact match for the generation is reported in below. For reference, the L2R free generation accuracy is 90.4%.

1. Regarding Forward X, R2L(m), free generation of $m$ using $\times n = p_{\text{correct}}$, the exact match accuracy for generative output is **82.1%**.

2. For Forward X, R2L(m), free generation of $m$ using $\times n = p_{\text{incorrect}}$, the generative exact match accuracy remains high at **30.2%**, while the accuracy for random guessing is nearly 0.

These results suggest that the R2L model may map unseen $p_{\text{incorrect}}$ to its neighborhood $p_{\text{correct}}$ and execute the same computation as if using $p_{\text{correct}}$. Consequently, the model faces a significant challenge in distinguishing alternative incorrect answers from the correct ones, making alternative incorrect answers hard-negatives.

# H    STATISTICAL TESTING RESULTS

We presents detailed statistical comparisons between L2R and R2L reasoning approaches across four tasks where R2L outperforms L2R: LogiQA, OpenBookQA, TruthfulQA, and CommonsenseQA. Metrics include mean accuracy, standard deviation, and the results of a paired t-test comparing the two approaches. All results were computed using bootstrap sampling with five replicates, each resampled at 80% with replacement, providing a robust estimation of performance variance. Notably, R2L consistently outperformed L2R, with statistically significant improvements ($p$-value $< 0.05$) observed across all tasks. The largest relative performance gain appeared in LogiQA, where R2L achieved a mean accuracy increase from 27.04% (L2R) to 32.56% (R2L) with an extremely significant $p$-value. OpenBookQA also exhibited a clear improvement, rising from 42.80% to 44.40%. TruthfulQA and CommonsenseQA saw meaningful gains as well, with $p$-values of 0.0072 and 0.0471, respectively, indicating reliable improvements beyond random fluctuation. Overall, these results underscore the effectiveness of reverse thinking (R2L) for improving reasoning tasks.

Table 8: Paired t-test of bootstrap samples (5 replicates, 80% resampled with replacement).

| Task | L2R Mean | L2R Std | R2L Mean | R2L Std | $p$-value |
|---|---|---|---|---|---|
| LogiQA | 27.04 | 0.00 | 32.56 | 0.0045 | 0.0000 |
| OpenBookQA | 42.80 | 0.00 | 44.40 | 0.0000 | 0.0000 |
| TruthfulQA | 25.08 | 1.16 | 28.93 | 0.80 | 0.0072 |
| CommonsenseQA | 42.17 | 1.34 | 44.68 | 1.70 | 0.0471 |