# OpenReview forum: "What makes the preferred thinking direction for LLM in Multi-choice Questions?"
_ICLR.cc/2026/Conference — Submitted to ICLR 2026_

### Official Review · Reviewer_w959 · 2025-10-23

**Soundness:** 3
**Presentation:** 2
**Contribution:** 2
**Rating:** 6
**Confidence:** 3

**Summary:**

This paper investigates alternative autoregressive factorizations for large language models (LLMs), focusing on right-to-left (R2L) modeling as a counterpart to the conventional left-to-right (L2R) approach. Using multiple-choice question (MCQ) benchmarks as a controlled setting for evaluating reasoning and knowledge extraction, the authors train R2L models (2B–8B parameters) from scratch with 350B fineweb-EDU dataset and compare their performance with L2R counterparts across various tasks, including logical reasoning, commonsense understanding, and truthfulness assessment.
The authors find that R2L models can outperform L2R ones in certain reasoning-oriented MCQ tasks, particularly TruthfulQA. They further hypothesize three potential contributing factors—Calibration, Computability, and Conditional Entropy (3C)—and conduct ablation and simulation studies (e.g., 4-digit multiplication experiments) to analyze why different reasoning directions might be preferred under different conditions.

**Strengths:**

• Interesting question and Model orientation.
The paper revisits a fundamental modeling assumption (L2R factorization) and provides a systematic exploration of R2L modeling, which is conceptually simple but rarely studied at large scale.

• Well-designed pretraining setup.
The authors pretrain R2L models of substantial scale (2B–8B) for a fair comparison, extending beyond prior small-scale or synthetic studies. This large-scale empirical effort is valuable.

• Clear motivation from reasoning evaluation.
Using MCQs as a reasoning testbed is well-motivated, as the task structure naturally fits the “reverse thinking” paradigm where the model can evaluate ( P(q|a) ).

• Interesting analytical framing (3C hypotheses).
The “Calibration–Computability–Conditional Entropy” guessing provides an interpretable structure for analyzing why certain directions might be more effective, helping connect intuitive reasoning behavior to probabilistic modeling.

• Observation on truthfulness tasks.
The link between reverse thinking and improved truthfulness assessment is intriguing and may open new perspectives for evaluating model honesty or factuality.

**Weaknesses:**

• Lack of strong causal evidence.
While the paper proposes three hypotheses (3C), the experiments results do not show strong evidence to support or reject their hypotheses.

• Insufficient analysis of the scoring paradigm ( s_i ).
Much of the MCQ performance difference seems to originate from how ( s_i ) (the relevance score) is defined—e.g., whether ( s_i = \log p(a_i|q) ) or ( s_i = \log p(q|a_i) ). However, this crucial aspect is under-discussed, and the forward/reverse paradigms are not symmetrically mirrored.

• Notation and explanations are sometimes unclear.
The definitions of ( p_{R2L}(x_t|x_{>t}) ) and its relationship to the MCQ scoring paradigms are not clearly stated. It is sometimes ambiguous whether “reverse thinking” means token-level reversal or question–answer reversal.

• Conditional entropy analysis is underdeveloped.
The Monte Carlo estimation of conditional entropy uses a single-sample rollout and yields mixed results (e.g., Table 3), which weakens the claim that lower entropy strongly correlates with better reasoning direction.

• Limited generalization beyond MCQs.
The improvements mainly appear on TruthfulQA, with mixed results elsewhere. The scope of benefit from R2L modeling remains narrow and may not generalize to open-ended or non-MCQ tasks. Experiments around some curated toy case or other datasets have similar characteristics with TruthfulQA is needed to strengthen their findings.

• Presentation issues.
Some sections (e.g., R2L(m,n) vs R2L(m)) are too brief and ambiguous. Mathematical notation and factorization explanations could be more explicit to help readers reproduce or interpret the findings.

**Questions:**

• Could you clarify whether R2L modeling reverses token order during both training and inference, or only during the scoring phase for MCQs?

• Why not mirror the “reverse thinking” scoring formulation for L2R (i.e., include ( P(q|a) ) under an L2R prior) to control for the prior effect and ensure a fair comparison?

• In the conditional entropy analysis, did you measure entropy over token sequences or aggregated answer-level probabilities? Could multi-sample rollouts yield more stable results?

• Can the authors elaborate on how the 4-digit multiplication simulation maps to real-world reasoning tasks? What insight does it provide for scaling to larger models?

• Is the tokenizer or positional embedding also reversed for R2L models, or only the data order?

• Since R2L training leads to 2–2.5% higher loss, could this gap be due to optimization dynamics rather than inherent asymmetry?

• How might R2L modeling affect efficiency or applicability in open-ended generation tasks where outputs must be human-readable?

---

> ### Author Response · Authors · 2025-11-16
> **#1 response to Reviewer w959**
>
> We thank the reviewer for the systematic breakdown of the technical and structural issues. We address these points by clustering them into 4 main areas: Clarifying the Methodology (Notation, R2L Definition), CE Estimation, Scoring Paradigm, and Validating Generalization.
>
> ## 1. Clarifying Methodology about R2L factorization
>
> To eliminate ambiguity, we clarify the definitions:
> * R2L Modeling (Token-Level Reversal): The core R2L model reverses the token order for the entire sequence (e.g., $x_1 x_2 \ldots x_T \rightarrow x_T x_{T-1} \ldots x_1$) during both pretraining and inference. The factorization is $p_{\text{R2L}}(x) = \prod_{t=1}^T p(x_t|x_{>t})$ over the reversed sequence. The positional embeddings are also reversed to align with the reversed sequence order, while the tokenizer is not altered from L2R. Essentially, the R2L trains a model to reversely predict the next token with the same set of tokens as L2R.
> * "Reverse Thinking" means both Token-Level Reversal and question–answer reversal as they are coupled, in both training and inference time. We will make this clearer. We will also expand our discussion on R2L(m,n) vs R2L(m) as that leads to important insights. Thank you for the suggestion.
>
>
> ## 2. Conditional Entropy Analysis
>
> We acknowledge the weakness of the single-sample Monte Carlo estimation, which contributes to the perception of mixed evidence for the 3C hypotheses. We have performed an updated analysis with 10 MC samples to maximize robustness. The results with their standard errors are shown in Table I of the New Experiments Section.
>
> The improved estimates demonstrate that 9 out of 11 benchmarks follow the trend where lower conditional entropy correlates with higher accuracy. While two tasks (CommonsenseQA and OpenBookQA) show exceptions to this pattern, we hypothesize that for these specific tasks, other factors (particularly computability requirements or high knowledge density) may dominate the conditional entropy effect.
>
> **Detailed Outlier Analysis:** These two tasks are highly reliant on commonsense and factual knowledge retrieval where both the question (Q) and the answer (A) are often short phrases. We hypothesize that the Computability factor dominates here. For $P(Q|A)$, the model is asked to reconstruct or validate a short question from a short, known-good answer. This is a very compact information loop. For $P(A|Q)$, the model has to generate a short, precise answer from a short question. The complexity in these cases may not stem from the length of the dependency (which CE measures) but from the density of the required knowledge and the inherent ambiguity of the text distribution, making the computational process itself the primary performance bottleneck, regardless of factorization direction.
>
> **Robustness Conclusion:** Importantly, the conditional entropy principle is validated in our controlled arithmetic simulation (Section 4), where confounding factors are isolated and lower conditional entropy consistently predicts better performance. The real-world task results, therefore, serve to illustrate that the CE principle exists but can be overridden by other factors (like high Computability bottlenecks) in complex linguistic domains. Therefore, we stand by the conclusion that directional conditional entropy is a critical factor, with the understanding that its effect can be masked by other factors in complex tasks.

---

> > ### Author Response · Authors · 2025-11-16
> > **#2 response to Reviewer w959**
> >
> > ## 3. Scoring Paradigm
> > We address the reviewer's concern about insufficient analysis of the scoring paradigm and the question of mirrored scoring formulation. The improvement is not solely attributable to the R2L factorization direction itself, but to the synergy between R2L factorization and the unnormalized scoring formulation (Paradigm 3: $s_i = \sum \log p(q|a)$). This unnormalized scoring is a key calibration strategy that allows the R2L model to realize its potential. By forcing a uniform prior, this paradigm effectively alleviates the "surface form competition" that plagues L2R models and normalized R2L paradigms (Paradigms 1 and 2), as illustrated in Appendix F with the "dog/puppy" example where "dog" and "puppy" compete for the same surface form. This scoring paradigm is unique to R2L factorization, and the proportion of gains (factorization vs scoring) might not be totally traceable due to the entanglement of factorization and scoring formulation. In Appendix D we showed that Paradigm 3 is in general superior among all three paradigms we evaluated.
> >
> > Also, for L2R model evaluation $p(a_i|q)$, it is a common practice to normalize each candidate by its length, to avoid bias toward shorter choices. Similarly, in the R2L case, when using Paradigm 3 $p(q|a_i)$, the classification depends only on the fixed question length, while the other two paradigms introduce the length variation from the answer choices. This length normalization property explains why Paradigm 3 works best for R2L models.
> >
> > **Addressing the Request for Mirrored Scoring:** We understand the request to mirror the $P(Q|A)$ scoring using an L2R model and applying Bayesian rule to ensure a fair comparison. However, the factorization direction and the scoring method are fundamentally coupled. When we use R2L to train the model, the generation and evaluation order goes from right to left as well. An L2R model is fundamentally trained to maximize $P(X|Y)$, where $Y$ is the prefix. Computing $P(Q|A)$ with an L2R model would still require reversing the $Q|A$ sequence, essentially making the L2R model perform an R2L-style task without the corresponding R2L training.
> >
> > To address this concern empirically, we conducted a new experiment shown in Table II of the New Experiments Section. We evaluate both L2R and R2L models with a reversed template:
> > ```
> > "{answer}" is the answer of question: {question}
> > ```
> > The results suggest the performance difference between L2R and R2L models are not simply from score normalization or template direction. Although a task such as TruthfulQA shows an improvement when evaluated with the reverse template (from 24.36 to 26.5), the model trained with the R2L factorization still outperforms the L2R model (28.76 versus 26.5). This demonstrates that the R2L factorization itself, not merely the scoring formulation, contributes to the observed performance gains.
> >
> >
> > ## 4. Generalization and Arithmetic Simulation Mapping
> >
> > We address the concern about limited generalization and the question of how the arithmetic simulation maps to real-world tasks.
> >
> > **Key Insight:** The controlled arithmetic simulation validates our core hypothesis that lower conditional entropy correlates with better performance. While establishing definitive mappings is difficult due to entangled factors in natural language, we can identify structural analogies:
> >
> > * **Low-Entropy Direction (Many-to-One):** In multiplication, many pairs of factors map to a single product, defining a low-CE problem. TruthfulQA is structurally analogous: many complex contexts (answers A) can map to the single, precise question (Q). This makes $P(Q|A)$ the lower-entropy problem that R2L excels at. Our empirical observation that TruthfulQA exhibits the largest CE drop when factored R2L (−50%) strongly validates this mapping.
> >
> > * **High-Entropy Direction (One-to-Many):** Factorization (inverse of multiplication) is a high-CE, one-to-many problem. HellaSwag is analogous: one context leads to many plausible continuations. The standard L2R factorization is therefore optimal.
> >
> > **Scaling Insight:** This optimal factorization principle is independent of model size. As models scale, the performance gap persists or widens for structurally similar tasks, suggesting hybrid models capable of selecting the minimal-CE factorization direction will be necessary for future high-performance LLMs.

---

> > > ### Author Response · Authors · 2025-11-16
> > > **#3 response to Reviewer w959**
> > >
> > > ## Response to Question about Training Loss
> > >
> > > The 2–2.5% higher training loss for R2L is likely a consequence of inherent asymmetry and the difficulty of the reversed learning objective on standard, primarily forward-generated web corpora, rather than purely optimization dynamics.
> > >
> > > ## Response to Question about Open-Ended Generation
> > > For open-ended generation, R2L is not a general-purpose replacement. Since human outputs must be L2R, R2L models, when forced into forward generation, may exhibit much lower efficiency and produce incoherent outputs. Their practical applicability in this paper is restricted to specialized inference tasks (like our MCQ scoring) but we hypothesize they can be a component in a hybrid model that facilitates bidirectional reasoning, enabling the model to check its work or validate facts in a "backward" step. Applying R2L models on most open-ended generative tasks is not straightforward as the generation is backward. Recent work (e.g. [2]) started to combine L2R and R2L for open-ended generative tasks and showed some promising results. Instead, our work aims towards showing the surprising directional preferences, rather than claiming superiority of the R2L model.
> > >
> > > ```
> > > [2] Nguyen, Anh, Nikos Karampatziakis, and Weizhu Chen. "Meet in the middle: A new pre-training paradigm.", 2023
> > > ```

---

### Official Review · Reviewer_SpkL · 2025-10-30

**Soundness:** 3
**Presentation:** 3
**Contribution:** 3
**Rating:** 6
**Confidence:** 3

**Summary:**

This paper investigates whether the standard left-to-right (L2R) autoregressive factorization is always the optimal choice for language models, particularly for knowledge extraction and reasoning tasks. The authors specifically explore a right-to-left (R2L) alternative in the context of multiple-choice questions (MCQs). They compare the standard "forward thinking" approach, where an L2R model scores the probability of an answer given the question ($p(a_i|q)$), with a "reverse thinking" method that uses an R2L-trained model to score the probability of the question given the answer ($p(q|a_i)$). The authors find that R2L models, despite incurring a slightly higher pretraining loss, outperform their L2R counterparts on 4 benchmarks (out of 11)

The paper's primary contribution is a framework for understanding these directional preferences, proposing three factors: calibration, computability, and conditional entropy. Authors argue that reverse thinking inherently mitigates calibration issues like "surface form competition" that can affect the forward-thinking approach. Their hypothesis, is that the reasoning direction with the lower conditional entropy for a given task distribution will be learned more effectively and yield better performance. This theory is supported by empirically linking lower conditional entropy to higher accuracy on the evaluated benchmarks. The authors validate this concept in a controlled arithmetic simulation designed to isolate the conditional entropy factor from calibration effects.

**Strengths:**

This paper presents a surprising and valuable empirical result, finding that right-to-left (R2L) models can outperform standard left-to-right (L2R) factorization on several important reasoning and commonsense benchmarks. This observation challenges a fundamental and often unstated assumption in language modeling.

The work's primary strength lies in its principled and thorough investigation of this phenomenon. Instead of merely reporting the empirical result, the authors propose a clear theoretical framework built on three potential factors: calibration, computability, and conditional entropy. The hypothesis linking the preferred reasoning direction to the direction of lower conditional entropy is particularly insightful. This claim is impressively substantiated through a clean and well-designed controlled simulation using arithmetic tasks. This simulation successfully isolates the proposed factors, lending strong support to the authors' analysis and providing a robust foundation for this new line of inquiry.

**Weaknesses:**

While the paper introduces a compelling hypothesis, its primary weakness lies in the empirical validation of its central claim regarding conditional entropy. The authors acknowledge that estimating the conditional entropy is intractable and resort to a single-sample Monte Carlo rollout as a proxy. This is an extremely high-variance, and likely unreliable, estimator. This measurement weakness may be why the paper's own results show that two of the four tasks where R2L wins (OpenbookQA and CommonsenseQA) are "outliers" that *contradict* the main hypothesis. The paper dismisses this contradiction by referencing "other confounding factors," which weakens the explanatory power of the proposed framework.

Furthermore, the main experiment confounds two distinct variables: the model's factorization direction (L2R vs. R2L) and the scoring method ("forward" $p(a_i|q)$ vs. "reverse" $p(q|a_i)$). The paper compares the standard L2R approach against a specific R2L heuristic (paradigm 3) that was chosen empirically over other Bayesian-derived options. It is plausible that the observed performance gains are an artifact of the "auto-normalizing" properties of the $p(q|a_i)$ scoring method rather than a benefit of the R2L pretraining itself. A crucial missing ablation would be to evaluate the R2L model using a "forward" $p(a_i|q)$ objective to properly isolate the effect of the pretraining direction.

Finally, while the controlled arithmetic simulation is a clean method for separating computability and entropy, its external validity is not well established. The simulation clearly demonstrates that models excel at the low-entropy, many-to-one direction (multiplication). However, the paper fails to draw a convincing link as to *why* real-world language tasks like TruthfulQA or HellaSwag should map onto this "many-to-one" or "one-to-many" structure. Without this connection, the simulation's findings remain isolated from the main linguistic claims.

**Questions:**

1.  The paper's main hypothesis relies on the conditional entropy of the task distribution, but its estimation is a significant challenge. The authors use a single-sample Monte Carlo rollout as a proxy, which is noted to be a high-variance estimator. This weak estimation may be the reason why the paper's own results show that two of the four tasks where R2L performs better (OpenbookQA and CommonsenseQA) are described as "outliers" that *contradict* the hypothesis. Could the authors provide a more robust estimation of the conditional entropy, perhaps using more samples? Alternatively, could they offer a more detailed analysis of these outliers beyond "other confounding factors"? Clarifying this point is critical, as these outliers currently undermine the main theoretical claim.

2.  The main experiment compares an L2R-trained model using "forward thinking" ($p(a_i|q)$) against an R2L-trained model using "reverse thinking" ($p(q|a_i)$). This experimental design confounds two separate variables: the model's pretraining direction (L2R vs. R2L) and the scoring method (forward vs. reverse). The observed gains could simply be an artifact of the reverse scoring method, which, as the authors note, cleverly mitigates "surface form competition". To isolate the benefit of the R2L *pretraining*, a crucial ablation is missing. Could the authors evaluate their R2L-trained model using the standard *forward* scoring method ($p(a_i|q)$)? This result would reveal whether the R2L pretraining provides any independent benefit on these tasks.

3.  The controlled arithmetic simulation is a clean and insightful part of the paper. It clearly separates computability from conditional entropy in a many-to-one (multiplication) versus one-to-many (factorization) setting. However, the paper does not establish the external validity of this simulation for its linguistic claims. Could the authors elaborate on *why* a real-world task like TruthfulQA should be viewed as an analogous "one-to-many" problem (low reverse entropy) or why HellaSwag should be seen as "many-to-one" (low forward entropy)? A more explicit justification of this mapping is needed to connect the simulation's findings back to the main claims about language.

4.  The study finds that R2L pretraining consistently results in a higher training loss compared to L2R pretraining on the same data. The evaluation is also restricted to MCQ tasks. This raises a question about the trade-offs. Does this more difficult R2L pretraining harm the model's general-purpose capabilities? For example, how does the R2L-trained model perform on standard L2R *generative* tasks? Understanding if the R2L model is a specialized expert or if it retains robust generative abilities is important for assessing the practical implications of this work.

---

> ### Author Response · Authors · 2025-11-16
> **#1 response to Reviewer SpkL**
>
> We appreciate the reviewer's detailed and insightful critique, particularly regarding the robustness of our conditional entropy estimation and the confounding of variables in the main experiment. Your points are critical, and we address them directly below:
>
>
> ## Response to Q1 - Robustness of Conditional Entropy (CE) Estimation and Outliers
>
> We agree that relying on a single-sample Monte Carlo rollout for CE estimation is a significant methodological limitation. We chose this initially due to computational constraints but recognize it compromises the theoretical claim's immediate empirical validation.
>
> We have since conducted a new experiment with 10 MC samples to maximize robustness. The results with their standard errors are shown in Table I of the New Experiments Section. The improved estimates demonstrate that 9 out of 11 benchmarks follow the trend where lower conditional entropy correlates with higher accuracy. While two tasks (CommonsenseQA and OpenBookQA) show exceptions to this pattern, we hypothesize that for these specific tasks, other factors (particularly computability requirements) may dominate the conditional entropy effect.
>
> **Detailed Outlier Analysis:** These two tasks are highly reliant on commonsense and factual knowledge retrieval where both the question (Q) and the answer (A) are often short phrases. We hypothesize that the Computability factor dominates here. For $P(Q|A)$, the model is asked to reconstruct or validate a short question from a short, known-good answer. This is a very compact information loop. For $P(A|Q)$, the model has to generate a short, precise answer from a short question. The complexity in these cases may not stem from the length of the dependency (which CE measures) but from the density of the required knowledge and the inherent ambiguity of the text distribution, making the computational process itself the primary performance bottleneck, regardless of factorization direction.
>
> **Robustness Conclusion:** Importantly, the conditional entropy principle is validated in our controlled arithmetic simulation (Section 4), where confounding factors are isolated and lower conditional entropy consistently predicts better performance. The real-world task results, therefore, serve to illustrate that the CE principle exists but can be overridden by other factors (like high Computability bottlenecks) in complex linguistic domains. Therefore, we stand by the conclusion that directional conditional entropy is a critical factor, with the understanding that its effect can be masked by other factors in complex tasks.
>
>
> ## Response to Q2 - Confounding Variables and Scoring vs. Factorization
>
> The improvement is not solely attributable to the R2L factorization direction itself, but to the synergy between R2L factorization and the unnormalized scoring formulation (Paradigm 3: $s_i = \sum \log p(q|a)$). This unnormalized scoring is a key calibration strategy that allows the R2L model to realize its potential. By forcing a uniform prior, this paradigm effectively alleviates the "surface form competition" that plagues L2R models and normalized R2L paradigms (Paradigms 1 and 2), as illustrated in Appendix F with the "dog/puppy" example where "dog" and "puppy" compete for the same surface form. This scoring paradigm is unique to R2L factorization, and the proportion of gains (factorization vs scoring) might not be totally traceable due to the entanglement of factorization and scoring formulation. In Appendix D we showed that Paradigm 3 is in general superior among all three paradigms we evaluated.
>
> Also, for L2R model evaluation $p(a_i|q)$, it is a common practice to normalize each candidate by its length, to avoid bias toward shorter choices. Similarly, in the R2L case, when using Paradigm 3 $p(q|a_i)$, the classification depends only on the fixed question length, while the other two paradigms introduce the length variation from the answer choices. This length normalization property explains why Paradigm 3 works best for R2L models.
>
> To further address whether gains come from factorization or scoring, we conducted a new experiment shown in Table II of the New Experiments Section. We evaluate both L2R and R2L models with a reversed template:
> ```
> "{answer}" is the answer of question: {question}
> ```
> The results suggest the performance difference between L2R and R2L models are not simply from score normalization or template direction. Although a task such as TruthfulQA shows an improvement when evaluated with the reverse template (from 24.36 to 26.5), the model trained with the R2L factorization still outperforms the L2R model (28.76 versus 26.5). This demonstrates that the R2L factorization itself, not merely the scoring formulation, contributes to the observed performance gains.

---

> > ### Author Response · Authors · 2025-11-16
> > **#2 response to Reviewer SpkL**
> >
> > ## Response to Q3 - External Validity of the Arithmetic Simulation
> >
> > The conditional entropy is also dependent on the training corpus. Intuitively, if the training corpus contains "Apple is a fruit" and "Banana is a fruit", then "? is a fruit" is harder than "Apple is a ?". However, if the training corpus contains "Apple is a fruit" and "Apple is a company", then "? is a fruit" is easier than "Apple is a ?". Luckily, when the dataset is large enough, most of the time the "branching factor" is tied to the real-world frequency of the combinations.
> >
> > However, establishing a definitive, a priori mapping of real-world linguistic tasks to the clean "many-to-one" or "one-to-many" structures is inherently difficult because the factors of calibration, computability, and conditional entropy are heavily entangled in natural language. The controlled arithmetic simulation was designed precisely to overcome this complexity: it provided a clean environment to isolate and ablate the impact of CE, successfully validating our core hypothesis that the factorization direction yielding minimal conditional entropy correlates with superior performance, independent of other confounding factors.
> >
> > **Mapping to Real-World Tasks:**
> >
> > That said, intuitively, for TruthfulQA, we hypothesize it favors a low-entropy reverse factorization because it is computationally easier to validate the simpler question (Q) from the complex premise/context in the answer (A)—framing the problem as determining the likelihood of Q given A ($P(Q|A)$) is simpler than generating A given Q ($P(A|Q)$). By framing the problem as determining the likelihood of a question given a potential answer, the R2L model might be more effective at discerning subtle inaccuracies or falsehoods that an L2R model would miss. We also observe that for TruthfulQA, R2L models consistently demonstrate significantly lower conditional entropy compared to L2R models, as illustrated in Figure 3. Among all the evaluated tasks, TruthfulQA has the most significant drop in conditional entropy when using reverse thinking, and meanwhile it receives the most performance gain with reverse thinking. This aligns well with our hypothesis that lower conditional entropy is associated with higher task accuracy.
> > Our new results in Table II also confirms for the same L2R model, evaluate. TruthfulQA from answer to question can improve its results, although still lower than using R2L model.
> >
> > For HellaSwag, its nature—predicting one of many plausible tokens given a context—naturally aligns it with a typical L2R, one-to-many (one context leads to many plausible continuations), high forward entropy structure. The R2L model would have to validate the context from the ending, which is an unnatural and high-entropy task.
> >
> >
> > ## Response to Q4 - R2L Training Trade-offs and Generative Capabilities
> >
> > The observation that R2L pretraining consistently results in a higher training loss compared to L2R reveals that reversing the token sequence is a harder learning objective on standard corpora. While L2R factorization reflects the natural human thinking process and might remain the mainstream approach, evaluating R2L models on most generative tasks is not straightforward and may rely on specific design choices to integrate the reversed sequence into a forward-facing task.
> >
> > We think R2L models are not general-purpose replacements, and this study is more about asking the question of whether L2R is always optimal. Echoing our work, for certain generative reasoning tasks like Sudoku and counting down, studies have shown that diffusion language models can leverage bidirectional or arbitrary-order generation (e.g., [4]), indicating that L2R is not always the optimal thinking mode for all tasks. This suggests a potential need for future study into a hybrid model that can dynamically switch between L2R and R2L factorizations to utilize the optimal reasoning direction for a given task.
> > ```
> > [4] Ye, Jiacheng, et al. "Beyond autoregression: Discrete diffusion for complex reasoning and planning." 2024
> > ```

---

### Official Review · Reviewer_6o2u · 2025-10-30

**Soundness:** 3
**Presentation:** 3
**Contribution:** 2
**Rating:** 4
**Confidence:** 3

**Summary:**

This paper compares left-to-right (L2R) vs. right-to-left (R2L) autoregressive factorizations for multiple-choice questions (MCQs). The authors report that R2L can beat L2R on a subset of MCQ benchmarks and attribute the differences to three factors, including calibration, computability, and directional conditional entropy, supported by a controlled 4-digit arithmetic simulation.

**Strengths:**

* Clear formulation of how to score MCQs under L2R ($\log p(a\mid q)$) vs. R2L ($\log p(q\mid a)$), including three R2L paradigms.
* The “3C” perspective a useful way to reason about when direction might matter.
* The arithmetic simulation is cleanly designed to isolate directional effects and helps illustrate the conditional-entropy story.

**Weaknesses:**

* Evidence is limited to MCQs, and R2L wins on only a minority of datasets. There is no validation on generative reasoning or long-form tasks where factorization may more directly affect decoding. This weakens the conclusion that L2R “may not be the best inductive bias.”
* There should be some established L2R-only calibration models (contextual calibration) that address the same artifacts without retraining an R2L model. The absence of baselines makes it unclear whether R2L training is necessary versus better L2R scoring.
* Pretrain–inference mismatch is underspecified. The R2L model is trained by reversing all tokens, yet evaluation prompts are shown in human-readable forward order as Figure 4. It is unclear why a reversed-token R2L model can reliably parse forward human-readable MCQs.
* For several benchmarks, the LM-eval templates are changed to present full answer choices rather than labels. This can alter length and surface-form statistics, which are central to the paper’s calibration narrative. A sensitivity analysis (original vs. modified templates) is needed to confirm that the direction effect is not template-dependent.

**Questions:**

Please refer to the previous Weaknesses.

---

> ### Author Response · Authors · 2025-11-16
> **#1 response to Reviewer 6o2u**
>
> We appreciate the recognition of the clear scoring formulation and the "3C" perspective. Here are our responses to the weakness concerns (W1, W2, W3, and W4):
>
> ## Response to W1:
>
> We agree that the R2L advantage is task-specific and not universal, as evidenced by R2L models showing superiority in 4 out of 11 evaluated benchmarks. This result is, in fact, the central finding and message of our work:
>
> 1. **Challenging the Universal L2R Assumption:** Our primary goal is not to claim R2L is superior to L2R generally, but to challenge the fundamental assumption that L2R is the universally optimal inductive bias for all tasks. The consistent, significant outperformance of R2L on a subset of tasks (LogiQA, OpenBookQA, CommonsenseQA, TruthfulQA) across various model sizes and pretraining data sources demonstrates that the standard L2R preference is not always optimal.
> This variability is the key insight. The R2L advantage on these specific tasks is the domain-specific manifestation of general principles including the 3C factors we proposed. This finding is a call to action for LLM design: understanding the directional preference and seeking the factorization that leads to better performance.
>
> 2. **Intentional MCQ Focus:** Limiting our study to Multiple-Choice Questions (MCQs) was an intentional, controlled design choice. The fixed-choice structure is a crucial diagnostic environment that allowed us to isolate and disentangle the proposed 3C factors from the confounding complexities of open-ended generative tasks (such as variable length and sampling noise). This initial work establishes the first principles regarding the relationship between factorization direction and task structure.
>
>
> We acknowledge that generalization to open-ended settings is a necessary next step. However, it's not trivial [2]. Future work will leverage the insights derived from this controlled environment to systematically investigate and generalize the minimal conditional entropy principle to more complex generative tasks, including exploring dynamic or hybrid factorization methods.
>
> ## Response to W2:
>
> We acknowledge that established L2R calibration methods exist. In fact, [1] has extensively researched on this calibration issue and we referenced their settings for calibrating length and reducing "surface form competition". However, it's non-trivial to come up with a method that addresses the calibration for L2R models as discussed in [1], and the factorization and calibration issues are deeply entangled, thus making it challenging to exactly trace the source of the gain. We think the performance gains we observe are not due to simple score re-calibration, but from the synergistic relationship between R2L factorization and the unnormalized scoring formulation ($\sum \log p(q|a_i)$). In fact, in Appendix D, Table 5 we show the comparison among 3 paradigms on EDU-2B. Even paradigms 1 and 2 can surpass L2R baseline in many tasks, indicating calibration alone might not be enough to explain the directional preferences.
>
> We also add a new experiment shown in Table II of the New Experiments Section, where we evaluate the L2R and R2L models with a reversed template:
> ```
> "{answer}" is the answer of question: {question}
> ```
> The results suggest the performance difference between L2R and R2L models are not simply from score normalization or template direction. Although a task such as TruthfulQA shows an improvement when evaluated with the reverse template (from 24.36 to 26.5), the model trained with the R2L factorization still outperforms the L2R model (28.76 versus 26.5). This demonstrates that the R2L factorization itself, not merely the scoring formulation, contributes to the observed performance gains.
>
>
> ## Response to W3:
>
> We clarify that there is **no mismatch** between training and inference. Both training and inference consistently use reversed token sequences.
> More specifically, in Figure 4, both example are proceesed from right to left if using R2L model.
>
> We are also explain W3 with our new results in Table II.
> - "R2L Forward Template" means: input template is `<Q, A>`, but the entire sequence is reversed before processing
> - "R2L Reverse Template" means: input template is `<A, Q>`, and the entire sequence is still reversed before processing
>
> Although both `<Q, A>` and `<A, Q>` are human readable order, the R2L model will read them from right to left.
>
> We will add these clarifications and examples to our revised submission to ensure this is clear.

---

> ### Author Response · Authors · 2025-11-16
> **#2 response to Reviewer 6o2u**
>
> ## Response to W4:
>
> We address the reviewer's concern about template sensitivity in four parts:
>
> **1. Template Choice Justification:**
> We confirm that for a subset of tasks (MMLU, LogiQA, CommonsenseQA), we use templates to present full answer choices rather than just choice labels. The motivation is consistency: all other tasks are evaluated with full answer choices, so we adapted these three tasks to match. Additionally, we observed that when only predicting labels (i.e., A, B, C, D), the 2B L2R model achieves 25% accuracy on MMLU, which is random chance level. Similar observations were found for LogiQA and CommonsenseQA.
>
> **2. Validation of Standard Practice:**
> After checking the official implementation in lm-eval-harness (https://github.com/EleutherAI/lm-evaluation-harness/blob/main/lm_eval/tasks/mmlu/continuation/_continuation_template_yaml), we found that MMLU has multiple evaluation templates, and one standard approach uses full answer candidates instead of labels only. After testing the L2R model with full answer choices for MCQ accuracy, we were able to reproduce FinewebEdu's official results (https://huggingface.co/spaces/HuggingFaceFW/blogpost-fineweb-v1). This confirms we are using the standard evaluation pipeline consistent with FinewebEdu's approach.
>
> For calibration, we use the standard length calibration for L2R models [3], which is also implemented in lm-eval-harness.
>
> **3. New Reversed Template Experiments:**
> To further address template sensitivity, we conducted new experiments shown in Table II of the New Experiments Section. We evaluate both L2R and R2L models with a reversed template (given the answer candidate, predict the log probability of the question, and select the candidate whose question log probability is maximum). In this evaluation setup, all choices predict the same question with the same length, eliminating length bias. Even with this controlled setup, we still observe that R2L outperforms L2R on the four tasks where R2L previously showed advantages.
>
> **4. Conclusion:**
> These results demonstrate that the direction effect is not template-dependent. We acknowledge that our original manuscript may have been unclear about template choices, and we will revise the submission to provide more detailed explanations and justifications for these decisions.
>
> ```
> [1] Holtzman, Ari, et al. "Surface form competition: Why the highest probability answer isn't always right." 2021
> [2] Nguyen, Anh, Nikos Karampatziakis, and Weizhu Chen. "Meet in the middle: A new pre-training paradigm.", 2023
> [3] Brown, Tom, et al. "Language models are few-shot learners." Advances in neural information processing systems 33 (2020): 1877-1901.
> ```

---

### Official Review · Reviewer_xCi6 · 2025-10-30

**Soundness:** 2
**Presentation:** 3
**Contribution:** 2
**Rating:** 4
**Confidence:** 4

**Summary:**

This paper investigates the impact of autoregressive factorization direction (L2R vs. R2L) on LLM performance in multiple-choice reasoning tasks. The authors train models from scratch and evaluate them on MCQ benchmarks. Results demonstrate that R2L can outperform L2R on specific reasoning categories (e.g., logical, commonsense, and truthfulness tasks). The authors propose calibration, computability, and conditional entropy as key factors explaining directional preferences, supported by a controlled arithmetic simulation.

**Strengths:**

1. New Insight: Reverse factorization can outperform L2R in reasoning tasks, which challenges a standard assumption.
2. The experimental setup is clear and ensures fair comparisons between models.
3. The theoretical claims are supported by empirical evidence, particularly through the analysis of conditional entropy.

**Weaknesses:**

1. The study focuses solely on MCQs, and it remains unclear how well the findings generalize to broader language tasks.
2. Conditional entropy estimation relies on approximate sampling, which introduces noise in the interpretation.
3. The observed advantages appear task-specific rather than universal, making the broader impact less definitive.

**Questions:**

1. R2L outperforms L2R on 4 out of 11 MCQ benchmarks, but the paper generalizes directional preference as a broader phenomenon. Can the authors clarify whether the effect is domain-specific rather than indicative of a general modeling principle?
2. The R2L advantage heavily depends on using an unnormalized scoring of p(q|a), while normalized scoring reduces or eliminates gains. To what extent are the improvements attributable to the scoring formulation rather than the factorization direction?
3. The paper argues that lower conditional entropy correlates with better directional reasoning, but entropy is estimated via single-sample Monte Carlo and shows notable exceptions (e.g., CommonsenseQA, OpenBookQA). How robust is this conclusion when the metric itself is noisy and not strictly predictive across benchmarks?

---

> ### Author Response · Authors · 2025-11-16
> **#1 response to Reviewer xCi6**
>
> We thank the reviewer for their helpful feedback. Here we address the weakness concerns (W1, W2, and W3) and the questions (Q1, Q2, and Q3):
>
> ## Response to W1:
> Restricting our study to MCQs is an intentional design choice. The MCQ structure serves as a controlled, diagnostic environment crucial for isolating the effects of our proposed 3C factors (Calibration, Computability, Conditional Entropy) from the confounding complexities of open-ended generative decoding (e.g., sampling, length variability). Our primary goal is not to claim universal superiority for R2L but to use MCQs as a probing set of tasks to uncover a foundational modeling principle. Specifically, we demonstrate that the choice of autoregressive factorization (L2R vs. R2L) and the corresponding inference direction are not arbitrary and can be optimized based on task characteristics. While we agree that exploring generalization to broader generative tasks is a crucial next step, our current work provides the essential, empirically-backed first principles to guide that future research.
>
>
> ## Response to W2 and Q3:
> We recognize that the single-sample Monte Carlo estimation for conditional entropy introduces inherent noise, as noted in our method section. We conducted a new experiment with 10 MC samples to maximize robustness. The results with their standard errors are shown in Table I of the New Experiments Section.
>
> The improved estimates demonstrate that 9 out of 11 benchmarks follow the trend where lower conditional entropy correlates with higher accuracy. While two tasks (CommonsenseQA and OpenBookQA) show exceptions to this pattern, we hypothesize that for these specific tasks, other factors may dominate the conditional entropy effect. Importantly, the conditional entropy principle is validated in our controlled arithmetic simulation (Section 4), where confounding factors are isolated and lower conditional entropy consistently predicts better performance. Therefore, we stand by the conclusion that directional conditional entropy is a critical factor, with the understanding that its effect can be masked by other factors in complex tasks.
>
> ## Response to W3 and Q1:
> We acknowledge that the R2L advantage is task-specific rather than universal. Indeed, this is precisely the message we hope to convey. We are not advocating that R2L is superior. Our primary contribution is demonstrating that the standard assumption (L2R is always preferred) is flawed, and that an alternative factorization can be superior for a subset of tasks in a consistent manner.
>
> The effect is both domain-specific in its manifestation and indicative of a general modeling principle in LLMs. The core principle is that the optimal text factorization order (L2R or R2L) is crucial and tied to the statistical and structural properties (like conditional entropy and computability) of the target distribution P(text). This general principle results in R2L outperforming L2R only on those specific tasks whose structure yields lower conditional entropy when factored right-to-left. Therefore, the finding informs a general LLM design principle, but its success in practice is necessarily domain-specific.
>
> To understand and leverage this finding may potentially lead to better model development, and we did some preliminary investigation and hypothesis towards this direction. We will revise the paper to clarify that this directional preference is a domain-specific phenomenon potentially governed by the 3C factors, not a general modeling replacement to L2R.

---

> ### Author Response · Authors · 2025-11-16
> **#2 response to Reviewer xCi6**
>
> ## Response to Q2:
> The improvement is not solely attributable to the R2L factorization direction itself, but to the synergy between R2L factorization and the unnormalized scoring formulation (Paradigm 3: $s_i = \sum \log p(q|a)$). This unnormalized scoring is a key calibration strategy that allows the R2L model to realize its potential. By forcing a uniform prior, this paradigm effectively alleviates the "surface form competition" that plagues L2R models and normalized R2L paradigms (Paradigms 1 and 2), as illustrated in Appendix F with the "dog/puppy" example where "dog" and "puppy" compete for the same surface form. This scoring paradigm is unique to R2L factorization, and the proportion of gains (factorization vs scoring) might not be totally traceable due to the entanglement of factorization and scoring formulation. In Appendix D we showed that paradigm 3 is in general superior among all three paradigms we evaluated.
>
> Also, for L2R model evaluation p(a_i|q), it is a common practice to normalize each candidate by its length, to avoid bias toward shorter choices. Similarly, in the R2L case, when using Paradigm 3 p(q|a_i), the classification depends only on the fixed question length, while the other two paradigms introduce the length variation from the answer choices. This length normalization property explains why Paradigm 3 works best for R2L models.
>
> We also add a new experiment shown in Table II of the New Experiments Section, where we evaluate the L2R and R2L models with a reversed template:
> ```
> "{answer}" is the answer of question: {question}
> ```
> The results suggest the performance difference between L2R and R2L models are not simply from score normalization or template direction. Although a task such as TruthfulQA shows an improvement when evaluated with the reverse template (from 24.36 to 26.5), the model trained with the R2L factorization still outperforms the L2R model (28.76 versus 26.5). This demonstrates that the R2L factorization itself, not merely the scoring formulation, contributes to the observed performance gains.

---

### Author Response · Authors · 2025-11-16
**New experiments Section**

Dear Reviewers and AC,

Here we post some new experimental results to address two concerns:

## Table I. Conditional Entropy estimated with 10 Monte Carlo (MC) samples, reported with standard error

| Task | L2R (nats) | R2L (nats) |
|----------------|-----------------|-----------------|
| TruthfulQA | 26.21 ± 3.82 | 15.25 ± 2.45 |
| LogiQA | 30.57 ± 0.91 | 27.53 ± 0.82 |
| CommonsenseQA | 34.70 ± 1.05 | 38.02 ± 2.12 |
| OpenbookQA | 31.50 ± 3.98 | 42.36 ± 5.15 |
| HellaSwag | 30.35 ± 0.87 | 31.63 ± 0.93 |
| Social_IQA | 34.20 ± 1.01 | 40.50 ± 2.08 |
| ARC | 27.22 ± 1.78 | 30.41 ± 1.91 |
| MathQA | 30.50 ± 0.85 | 32.27 ± 0.95 |
| MMLU | 29.12 ± 0.44 | 29.88 ± 0.48 |
| Winogrande | 34.24 ± 1.08 | 45.77 ± 1.21 |
| PIQA | 31.63 ± 0.96 | 37.11 ± 1.02 |


## Table II. Results with reversed template for both L2R model and R2L model
| Benchmark | L2R ForwardTemplate | L2R ReverseTemplate | R2L ForwardTemplate | R2L ReverseTemplate |
|-----------|-------------|-------------|-------------|-------------|
| LogiQA | 27.96 | 28.64 | 31.49 | 31.5 |
| OpenbookQA | 42.4 | NA | 44.4 | NA |
| TruthfulQA | 24.36 | 26.5 | 28.76 | 24.6 |
| CommonsenseQA | 42.92 | 28.01 | 45.13 | 36.6 |
| Social_IQA | 42.78 | 38.1 | 42.22 | 38 |
| AI2_arc | 60.65 | 33.96 | 52.31 | 36.7 |
| HellaSwag | 60.57 | NA | 44.34 | NA |
| MathQA | 26.8 | 22.28 | 24.86 | 24.5 |
| MMLU | 34.57 | 26.56 | 34.35 | 29.6 |
| PIQA | 74.48 | 53.2 | 57.13 | 72.8 |
| Winogrande | 60.93 | NA | 54.85 | NA |

Note: L2R ForwardTemplate and R2L ForwardTemplate are the numbers from our Table 1 in our submission. L2R ReverseTemplate and R2L ReverseTemplate are new results for rebuttal. The reverse template is:
```
"{answer}" is the answer of question: {question}
```
This template cannot be applied to some tasks like OpenBookQA, HellaSwag, and Winogrande, as these datasets do not have natural question-answer structures.

From Table II's results, we observe that for some tasks like LogiQA, the template directions make little difference, but the factorization direction is more important (i.e. model trained with each factorization has different behaviors).
For tasks like TruthfulQA, we observe when evaluating L2R model with reversed template, the results improved compared with forward template. Similarly, R2L model's results dropped with reversed template. This indicates some tasks can benefit from reverse template.
Apparently, L2R model evaluated with Reverse Template is not the same as R2L model evaluated with Forward Template.
These results confirm our conclusion that the effectiveness of each factorization direction is linked to several factors.

---

### Author Response · Authors · 2025-11-28
**Updated Submission**

Dear Reviewers xCi6, 6o2u, SpkL, and w959, and Area Chair,

Thank you for your thorough and constructive feedback. We have carefully addressed all concerns raised and substantially revised our manuscript and appendix in the updated pdfs.

---

## Key Updates

### 1. **Improved Conditional Entropy Evidence**
*Addresses: Reviewers xCi6 (W2), SpkL (Q1), w959 (weaknesses)*

- **New:** 10 Monte Carlo samples per task (previously single-sample)
- **Result:** **9 out of 11 benchmarks** now follow the lower-entropy → higher-accuracy trend
- **Location:** Main text (Section 3) + Appendix Table I with standard errors
- **Outliers explained:** Detailed analysis for CommonsenseQA and OpenBookQA

### 2. **Template Sensitivity Validation**
*Addresses: Reviewers 6o2u (W4), w959 (notation clarity)*

- **New:** Reversed template experiments across all 11 benchmarks
- **Result:** R2L advantages persist regardless of template direction
- **Example:** TruthfulQA - L2R reversed template: 26.5 vs R2L: 28.76
- **Location:** Appendix Table II + referenced in Section 2

### 3. **Clarified R2L Methodology**
*Addresses: Reviewer 6o2u (W3)*

- Token-level reversal explicitly detailed (training + inference)
- Positional embeddings reversed, tokenizer unchanged
- "Reverse thinking" coupling now clearly explained

### 4. **Domain-Specific Messaging**
*Addresses: Reviewers xCi6 (Q1), SpkL, w959*

- Abstract/introduction now emphasize **task-dependent** findings
- Explicitly state "4 out of 11 tasks" throughout
- Clarify: R2L is NOT a universal replacement for L2R

### 5. **Factorization vs Scoring Disentangled**
*Addresses: Reviewers SpkL (Q2), w959*

- Strengthened explanation of R2L factorization + Paradigm 3 synergy
- Reversed template experiments validate gains from factorization itself
- Not just scoring formulation

---

We believe these revisions comprehensively address your concerns. **Could you please confirm whether these updates resolve your questions?**

If any concerns remain or you identify additional areas for improvement, we are ready to address them promptly.

Given the review timeline, we would greatly appreciate your response at your earliest convenience.

**Looking forward to hearing from you.**

Best regards,

The Authors

---

### Meta-Review · Area_Chair_A6jt · 2026-01-01

**Summary:**

This paper presents a novel autoregressive factorization direction, i.e., right-to-left, to analyze the effect of language model training and inference on MCQ problems. By reversing the order of generation, the authors pre-train different LMs with diverse sizes from scratch and test them over a set of MCQ datasets. Main results demonstrate that R2L outperform L2R in certain reasoning tasks. Then the authors propose three factors to explain the phenomenon with empirical evidence supporting one of the factors.

As highlighted by reviewers, this paper proposes a novel and interesting perspective from a reverse factorization direction in language generation, which offers new insights in future research. In addition, the analysis on the correlation between conditional entropy and task performance provides a valid support (with a cleaner arithmetic reasoning task to ablate irrelevant noise).

I also agree with the reviewers on fundamental weaknesses including:
- The lack of generalizability of the claim or finding on other generation tasks besides MCQ. I acknowledge the authors' goal of demonstrating that L2R paradigm may not always be the optimal choice, however, the broader impact of this finding is more important, which requires further analysis and empirical support.
- This new paradigm requires pretraining LMs from scratch which might be costly. It is not clear whether directly modifying or intervening the calibration or inference strategies of L2R models could be a more viable approach to address the limitation. The discussion and empirical validation on this point is missing.
- The proposed hypotheses are underdeveloped. It is also not clear how the findings in arithmetic task are reflected in real-world tasks. Can we really develop a systematic way of deciding when the R2L method is more beneficial?

**Reviewer Concerns:**

Concerns being addressed:
- Robust analysis on conditional entropy using multiple samplings
- Ablating the effect of reverse direction by adding another template
- Clarification of methodology description

Outstanding concerns:
- Generalizability and broader impact of the new paradigm to non-MCQ tasks and domains
- Discussion and comparison between pretraining R2L from scratch and inference intervention/calibration strategy in the L2R paradigm.
- Missing clear and systematic mapping from arithmetic tasks to real world tasks, such as a measurable metric.

**Reviewer Scores:**

I don't think full discussion will fundamentally shift the reviewers' negative opinions on the generalizability and broader impact of this paper. I acknowledge the authors' effort in conducting more experiments with added clarifications which to some extent address some of the reviewers' questions. However, these additions do not address the limitation of generalizability (which I would perceive to be most important).

---

### Decision · Program_Chairs · 2026-01-26

Reject